# Feasibility-Driven Trust Region Bayesian Optimization

**Paolo Ascia**[1]  **Elena Raponi**[2]  **Thomas Bäck**[2]  **Fabian Duddeck**[1]

[1]Technical University of Munich, Arcisstr. 21, 80333 Munich, Germany
[2]LIACS, Leiden University, Einsteinweg 55, 2333 CC Leiden, The Netherlands

**Abstract**     Bayesian optimization is a powerful tool for solving real-world optimization tasks under tight evaluation budgets, making it well-suited for applications involving costly simulations or experiments. However, many of these tasks are also characterized by the presence of expensive constraints whose analytical formulation is unknown and often defined in high-dimensional spaces where feasible regions are small, irregular, and difficult to identify. In such cases, a substantial portion of the optimization budget may be spent just trying to locate the first feasible solution, limiting the effectiveness of existing methods. In this work, we present a Feasibility-Driven Trust Region Bayesian Optimization (FuRBO) algorithm. FuRBO iteratively defines a trust region from which the next candidate solution is selected, using information from both the objective and constraint surrogate models. Our adaptive strategy allows the trust region to shift and resize significantly between iterations, enabling the optimizer to rapidly refocus its search and consistently accelerate the discovery of feasible and good-quality solutions. We empirically demonstrate the effectiveness of FuRBO through extensive testing on the full BBOB-constrained COCO benchmark suite and other physics-inspired benchmarks, comparing it against state-of-the-art baselines for constrained black-box optimization across varying levels of constraint severity and problem dimensionalities ranging from 2 to 60.

## 1 Introduction

The global optimization of black-box objective functions under expensive, black-box constraints—where both are only accessible via costly point-wise evaluations—is a fundamental problem in fields such as machine learning (ML), engineering design, robotics, and natural sciences. For instance, in automated machine learning [Hutter et al., 2019], black-box optimization techniques, and in particular Bayesian optimization (BO) [Garnett, 2023], are commonly used to tune hyperparameters of ML models to maximize predictive performance under strict constraints on model inference time, memory footprint, or energy consumption. This setup is common in frameworks like Auto-sklearn [Feurer et al., 2022], AutoKeras [Jin et al., 2023], or custom pipelines for neural architecture search under deployment constraints [Cai et al., 2019]. Constrained BO is also widely used in crashworthiness optimization [Raponi et al., 2019, Du et al., 2023] to efficiently tune design parameters for objectives like weight or energy absorption, under constraints such as intrusion depth or peak acceleration. In these settings, evaluating either the objective or constraints can be costly and time-consuming, often relying on physical experiments or computationally intensive simulations.

The challenge of efficiently addressing black-box constrained problems is further amplified in high-dimensional settings [Powell, 2019], meaning problem settings with dozens of decision variables in the context of BO. In fact, as the volume of the search space increases, sampling becomes sparse, surrogate models like Gaussian process regression models become harder to fit due to the reduced correlation between points, optimization landscapes become more complex, with many local optima and constraint boundaries that are trickier to approximate, and feasible regions become narrow and non-convex islands in a vast space. A large portion of evaluations may land in infeasible zones, and even identifying a single feasible point may consume a large portion—or even all—of the available evaluation budget. This renders many existing BO methods ineffective.

**Our contribution.** Our work builds directly upon the Scalable Constrained Bayesian Optimization (SCBO) algorithm [Eriksson and Poloczek, 2021], which introduced a scalable trust-region-based framework for constrained BO in high-dimensional settings. SCBO demonstrated that localizing the search using dynamically-adapted trust regions, rather than relying on global surrogate optimization, offers both scalability and performance benefits. However, the trust regions defined by SCBO make only partial use of the information deriving from the modeling of the problem constraints, specifically to center the trust region and select the next candidate solutions, demonstrating particular effectiveness in the case where feasible regions are relatively easy to find—a condition that often does not hold in the most challenging constrained problems. We hence propose the *Feasibility-Driven Trust Region Bayesian Optimization* (FuRBO) algorithm, specifically designed to tackle high-dimensional constrained optimization problems where finding any feasible point is itself difficult. FuRBO retains the core idea of adaptive trust regions but shifts the focus to feasibility-first exploration relying on the constraint isocontour predicted by the surrogate model. To construct the trust region, FuRBO leverages both the objective and constraint surrogate models. At each iteration, FuRBO samples a set of points—referred to as *inspectors*—uniformly distributed within a ball of radius $R$ centered at the best candidate found so far. These inspectors are evaluated over the constraint landscape and used to estimate the likely location and shape of the feasible region. The most promising inspectors, ranked using both the objective and constraint models, determine the position, shape, and size of the trust region for the next search step. Within this feasibility-guided trust region, it then applies Thompson sampling on the objective and constraint models to identify new promising points to query. Through a series of comprehensive experiments on the full BBOB-constrained COCO benchmark suite [Dufossé et al., 2022] and other physics-inspired benchmarks, both containing problems with increasing constraint complexity, we show that FuRBO, thanks to its landscape-aware mechanism that uses inspector sampling to guide the search toward promising feasible regions, either ties or outperforms other state-of-the-art alternatives for constrained BO, with evident superiority in settings in which feasibility is rare and hence difficult to locate.

**Reproducibility**: The code for reproducing our experiments, along with the whole set of figures, is available on GitHub[1].

## 2 Related work

Bayesian optimization (BO) [Garnett, 2023] is a sample-efficient, model-based optimization framework for solving expensive black-box problems where function evaluations are costly or time-consuming. On continuous search spaces, a Gaussian Process (GP) is commonly used in BO to define a prior distribution over the unknown objective function, capturing assumptions about its smoothness and variability. The process begins with an initial set of evaluated points (also known as Design of Experiments [Forrester et al., 2008]) typically selected through random sampling or space-filling designs. Once data from the initial evaluations is available, the GP is conditioned on these observations to yield a posterior distribution, which provides an approximation of the unknown objective function along with uncertainty estimates. An acquisition function (AF) is then used to decide where to evaluate next by balancing exploration (sampling in regions of high uncertainty) and exploitation (sampling where high objective values are likely). This iterative process continues until the evaluation budget is exhausted or convergence is reached.

Constrained BO extends the classical BO framework to settings where one must optimize an objective function subject to one or more unknown or expensive-to-evaluate constraints. This is common in real-world scenarios, such as engineering design or hyperparameter tuning, where feasible solutions must satisfy safety, performance, or resource limits. Despite most work on BO has focused on unconstrained scenarios, some extensions to constrained optimization problems have been introduced in the last years. The constrained expected improvement (CEI), introduced

---

[1] https://github.com/paoloascia/FuRBO

by Schonlau et al. [1998] and popularized by Gardner et al. [2014], is the earliest and most widely used technique for handling constraints in BO. It extends the standard Expected Improvement (EI) AF to handle constraints by multiplying the improvement with the probability that a candidate solution is feasible. This allows the algorithm to prioritize sampling in regions that are not only promising in terms of objective value but also likely to satisfy the given constraints.

The Predictive Entropy Search with Constraints (PESC) AF by Hernández-Lobato et al. [2016] focuses in particular on problems with decoupled constraints, in which subsets of the objective and constraint functions may be evaluated independently. It extends the entropy search AF by not only reducing uncertainty about the location of the global optimum, but doing so under the constraint that the solution must also be feasible.

Picheny et al. [2016] proposed SLACK, which augments the standard constrained Bayesian optimization framework with slack variables to reformulate equality constraints as inequalities. By combining this with an augmented Lagrangian approach and EI, they demonstrated improved performance in problems with equality constraints.

Ariafar et al. [2019] advanced the augmented Lagrangian framework by integrating the Alternating Direction Method of Multipliers (ADMM), allowing a more scalable and structured optimization of constrained black-box problems. Their method also uses EI to select query points and is particularly suited to problems with multiple and decoupled constraints.

Ungredda and Branke [2024] proposed a variant of the Knowledge Gradient (KG) AF—called the constrained Knowledge Gradient (cKG)—to handle constrained optimization problems. In cKG, feasibility is incorporated into the Bayesian lookahead by weighting the expected utility from the objective GP with the estimated probability of feasibility from the constraint GPs, guiding the search toward points that are both promising and likely to be feasible.

All of these methods were not designed with high-dimensional problems in mind and often struggle with scalability. This limitation was instead addressed in the design of the Scalable Constrained Bayesian Optimization (SCBO) framework by Eriksson et al. [2019], which introduced a surrogate-based framework that models the objective and each constraint separately, allowing for greater flexibility and modularity in the modeling process. It uses trust regions as a core component, using them to search for new candidate solutions locally, in regions with predicted high feasibility and optimality, allowing for robust scalability to high-dimensional constrained spaces. Despite the introduction of new methods in recent years (see the survey by Amini et al. [2025] for a comprehensive overview), SCBO remains a state-of-the-art approach for constrained high-dimensional BO. Unlike most methods reviewed, despite SCBO being developed in response to practical challenges, its performance has been rigorously benchmarked also on standard test problems. This has contributed to its robustness, establishing SCBO as a standalone optimization framework that is also accessible through the well-known `BoTorch` [Balandat et al., 2020] package. For this reason, we developed our method, FuRBO, building on SCBO as a foundation, but redefining the trust region design procedure to more effectively address problems with narrow, hard-to-find feasible regions.

## 3 Problem definition

We consider the problem of minimizing a black-box objective function $f : \Omega \rightarrow \mathbb{R}$ subject to multiple constraints. The goal is to identify an optimal design point $x^* \in \Omega \subset \mathbb{R}^D$ that maximizes the objective while satisfying all constraints:

$$x^* = \arg\min_{x \in \Omega} \quad f(x)$$
$$\text{subject to} \quad c_k(x) \leq 0, \quad \forall k \in \{1, \ldots, K\} \tag{1}$$

Alongside the objective, the constraint functions $c_k : \Omega \rightarrow \mathbb{R}$ return a vector $\mathbf{c}(x) = [c_1(x), \ldots, c_K(x)]$ that quantifies the feasibility of a sample. A point is considered to be feasible if it belongs to the set $\Omega_{\text{feas}} = \{x \in \Omega \mid c_k(x) \leq 0 \; \forall k \in \{1, \ldots, K\}\}$.

We assume a limited evaluation budget of $10D$ function evaluations, reflecting the practical setting of real-world applications where each evaluation is costly and only a small number of queries is affordable. This low-budget scenario is precisely where BO methods are most effective. After using the total evaluation budget, the algorithm recommends a solution $x_{\text{best}} \in \Omega$. If $x_{\text{best}} \in \mathcal{F}$, we measure the quality of a recommendation by loss, i.e., the simple regret, under feasibility conditions: $l(x_{\text{best}}) = f(x_{\text{best}}) - f(x^*)$, where $x^*$ is the global optimum of the problem. If $x_r \notin \mathcal{F}$, the solution is considered infeasible and its maximum constraint violation ($V_{\max}(x) = \max_{i=1,\ldots,K} \max\{0, c_k(x)\}$) is returned instead.

## 4 Feasibility-Driven Trust Region Bayesian Optimization

To overcome some of the limitations of optimizing highly constrained problems, we propose a new algorithm: FuRBO. Our method shares the idea of using trust regions for BO introduced by Eriksson et al. [2019]. However, instead of using the best-evaluate sample to define only the center of the trust region, we identify its position and extension using the information available from the approximation models of both the objective and constraint functions. What distinguishes our approach from SCBO is the formulation of the trust region. We therefore begin by outlining the SCBO framework, followed by a detailed explanation of how the trust region is defined in FuRBO.

### 4.1 SCBO algorithm

The SCBO framework extends the TuRBO algorithm [Eriksson et al., 2019] to address problems with black-box constraints, by preserving most of the algorithm structure. It begins by evaluating an initial design and fitting Gaussian Process (GP) models to the objective $f(x)$ and constraints $c_k(x)$ for $k = 1, \ldots, K$. A trust region (TR) is initialized around the best feasible point; if none is found, it is centered at the point with the smallest constraint violation.

The algorithm then iteratively proceeds until the evaluation budget is exhausted. In each iteration, a batch of $q$ candidate points is identified within the current trust region using a Thompson Sampling (TS) [Thompson, 1933] AF: to return each of the $q$ points, after sampling a large set of $r$ candidate solutions within the TR, a realization $\{(\hat{f}(x_i), \hat{c}_1(x_i), \ldots, \hat{c}_K(x_i)) \mid 1 \leq i \leq r\}$ from the posterior of both objective and constraints is sampled, and the candidate with maximum utility among those that are predicted to be feasible is added to the batch.

Once the batch of $q$ points is selected, the algorithm evaluates both the objective and constraint functions at these locations. The TR is then updated: its center is moved to the best feasible point found so far, some success/failure counters ($n_s, n_f$) are updated, and the TR size $L$ (same for all dimensions) is adjusted if either $n_s$ or $n_f$ reach some threshold for the update, $\tau_s$ and $\tau_f$, respectively. If $L$ becomes smaller than a predefined threshold $L_{\min}$, the whole procedure is reinitialized.

At the end of the optimization, SCBO recommends the best feasible point found, i.e., the point with the smallest objective value among those satisfying all constraints $c_k(x) \leq 0$, for $k = 1, \ldots, K$. We point the reader to the original paper by Eriksson and Poloczek [2021] for more details.

### 4.2 FuRBO algorithm

The main novelty of FuRBO is the definition of the TR. The other optimization steps are shared with SCBO [Eriksson and Poloczek, 2021]. Nevertheless, for the sake of completeness, we present in this section the entire optimization flow. We provide an illustration of the update procedure for the TR in Figure 1 and the pseudocode of the entire FuRBO framework in Algorithm 1, with the lines that differ from SCBO highlighted in yellow for clarity.

As shown in line 1 of Algorithm 1, FuRBO begins by sampling the entire search space $\Omega$, following the standard initialization procedure of vanilla BO. It then evaluates the sample points

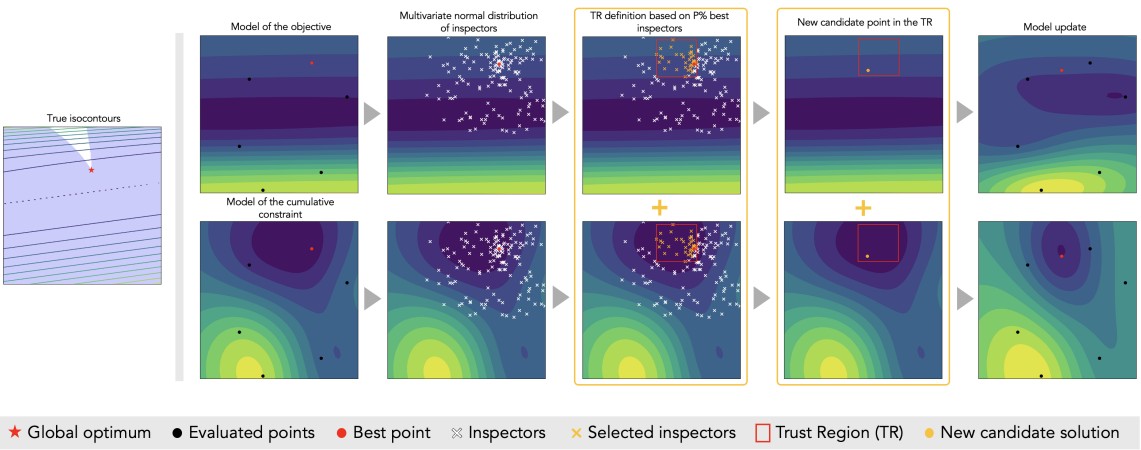

Figure 1: One iteration of FuRBO. The leftmost panel shows the true objective and constraint isocontours, with the global optimum in red. The next two panels show surrogate models of the objective (top) and aggregated constraint (bottom), built from evaluated points (black dots); the current best solution is marked in red. Inspectors (white crosses) are sampled around this point and ranked by feasibility and objective value. The top $P_\%$ (orange crosses) define the TR (red square), using both objective and constraint models. A new candidate (orange dot) is proposed within the TR, and the model are updated after evaluation (rightmost panel).

on the true objective and constraint functions, hence generating a set $\mathcal{S} = \{(x_i, f(x_i), \mathbf{c}(x_i))\}_{i=1}^N$ of evaluated points. Here, $\mathbf{c}(x_i) = [c_1(x_i), \ldots, c_K(x_i)]$ is a vector with as many components as the number of constraints defining the problem. Let $\mathcal{F} = \{x_i \in \mathcal{S} \mid c_k(x_i) \leq 0, \forall k = 1, \ldots, K\}$ be the feasible set, i.e., the subset of feasible points in $\mathcal{S}$ according to all the constraints. Let $\bar{\mathcal{F}} = \mathcal{S} \setminus \mathcal{F}$ be its complement. At each iteration of the optimization procedure, until the evaluation budget is exhausted, the following steps are performed.

**Rank samples**. For all feasible samples $x_i \in \mathcal{F}$, define the ranking by the true objective value $f(x_i)$ (lower is better for minimization). Let the feasible samples be sorted such that $f(x_1^{\text{feas}}) \leq f(x_2^{\text{feas}}) \leq \cdots \leq f(x_{|\mathcal{F}|}^{\text{feas}})$. For the infeasible samples, we first normalize each constraint dimension over all infeasible points: $\tilde{c}_k(x_i) = \frac{c_k(x_i)}{\max |c_k(x_i)|}$ for $x_i \in \bar{\mathcal{F}}$, $k = 1, \ldots, K$. We adopt this definition of normalization to preserve the boundary between feasibility and infeasibility. We then define the maximum normalized constraint violation per sample $v(x_i) = \max_{k=1,\ldots,K} \tilde{c}_k(x_i)$, and rank infeasible samples by ascending $v(x_i)$ (smallest violation first): $v(x_1^{\text{infeas}}) \leq v(x_2^{\text{infeas}}) \leq \cdots \leq v(x_{|\bar{\mathcal{F}}|}^{\text{infeas}})$. Finally we concatenate the ordered feasible and infeasible samples in $\mathcal{S}_{\text{ranked}} = \left[ x_1^{\text{feas}}, \ldots, x_{|\mathcal{F}|}^{\text{feas}}, x_1^{\text{infeas}}, \ldots, x_{|\bar{\mathcal{F}}|}^{\text{infeas}} \right]$. The procedure described in this step is what defines the ranking metric $r$ in Algorithm 1.

**Generate the inspectors**. We select the top-ranked point as the best candidate solution $x_{best}$ so far (line 2). Around this point, we sample a set of inspector points $\mathcal{I} = \{x_1, \ldots, x_N\}$ by first drawing from a multivariate normal distribution $\mathcal{N}(0, \sigma^2 I_D)$, normalizing each sample to lie on the unit hypersphere, then scaling by a random factor in $[0, R]$ and finally translating by $x_{\text{best}}$, ensuring the inspectors are uniformly distributed within a ball $\mathcal{B}(x_{\text{best}}, R)$ of radius $R$ centered at $x_{\text{best}}$ (line 4).

**Definition of the TR**. The inspector population in $\mathcal{I}$ is ranked as described in Step (1), but based on the evaluated models $\mathcal{M}$ and $\{\mathcal{C}_k, \forall k = 1, \ldots, K\}$ of the objective and $K$ constraints. We hence define the ranked list $\mathcal{R}$ over $\mathcal{I}$ as $\mathcal{R} = \mathtt{rank}(\mathcal{I}; \mathcal{M}, \mathcal{C}_i) = \mathtt{sorted}(\mathcal{I}, \text{by increasing } r(x))$ (line 5). Then, we select the top $P_\%$ inspectors: $\mathcal{I}_{\text{best}} = \{x \in \mathcal{R} \mid \mathtt{rank}(x) \leq \lceil P \cdot N \rceil\}$ (line 6) and define the

**Algorithm 1** FuRBO algorithm

---

**Require**: Success threshold $\tau_s$, failure threshold $\tau_f$, success counter $n_s$, failure counter $n_f$, batch size $q$, inspector percentage $P_\%$, sample set $\mathcal{S} = \emptyset$, surrogate model of the objective $\mathcal{M}$, surrogate model of the constraints $\mathcal{C}_k$, sampling radius $R$, search space $\Omega$, initial trust region TR $= \Omega$, Thompson sampling AF TS, function to optimize $f$, constraint functions $c_k$, ranking metric $r$

1: Evaluate initial design, update $\mathcal{S}$ and train surrogate models $(\mathcal{M}, \mathcal{C}_k)$
2: $x_{\text{best}} \leftarrow \arg\min_{x \in X} r(x; f, c_k)$         ▷ Update best candidate solution
3: **while** Optimization Budget Not Exhausted **do**
4:      $\mathcal{I} \leftarrow \text{UniformBallSamples}(x_{\text{best}}, R)$    ▷ Sample inspectors uniformly within $\mathcal{B}(x_{\text{best}}, R)$
5:      $\mathcal{R} \leftarrow \text{rank}(\mathcal{I}; \mathcal{M}, \mathcal{C}_k)$           ▷ Rank inspectors based on $(\mathcal{M}, \mathcal{C}_k)$
6:      $\mathcal{I}_{\text{best}} \leftarrow$ Top $P_\%$ of sorted $\mathcal{I}$      ▷ Select the best inspectors ranked according to $\mathcal{R}$
7:      TR $\leftarrow \text{define\_TR}(\mathcal{I}_{\text{best}})$    ▷ Define TR as the smallest hyperrectangle containing $\mathcal{I}_{\text{best}}$
8:      $X_{\text{next}} \leftarrow TS((\mathcal{M}, \mathcal{C}_k), \text{TR}, q)$    ▷ Propose next $q$ configurations to evaluate within the TR
9:      $Y \leftarrow f(X_{\text{next}})$           ▷ Evaluate objective function on the new points
10:     $C_k \leftarrow c_k(X_{\text{next}})$         ▷ Evaluate constraint functions on the new points
11:     $\mathcal{S} \leftarrow \mathcal{S} \cup \{(X_{\text{next}}, Y, C_k)\}$          ▷ Update sample set
12:     Fit surrogate models $(\mathcal{M}, \mathcal{C}_k)$ over $\Omega$
13:     Update $n_s$ and $n_f$
14:     **if** $n_s = \tau_s$ or $n_f = \tau_f$ **then**       ▷ Check if thresholds for radius update is reached
15:        $R \leftarrow \text{adjust}(R)$        ▷ Double/halve the variance of the distribution
16:     **end if**
17:     $x_{\text{best}} \leftarrow \arg\min_{x \in X} r(x; f, c_k)$         ▷ Update best candidate solution
18: **end while**
19: Return $x_{\text{best}}$               ▷ Return best solution

---

TR as the smallest hyperrectangle that contains all points in $\mathcal{I}_{\text{best}}$. Let $x_j^{\min} = \min_{x \in \mathcal{I}_{\text{best}}} x_j$ and $x_j^{\max} = \max_{x \in \mathcal{I}_{\text{best}}} x_j$, for $j = 1, \ldots, d$, then TR $= \prod_{j=1}^{d} [x_j^{\min}, x_j^{\max}]$ (line 7).

**Find new candidate solutions, update sample set and posterior distributions**. Following the SCBO algorithm, we use TS over the surrogate models $(\mathcal{M}, \mathcal{C}_k)$, restricted to the current TR, to propose a batch of $q$ new points to evaluate (line 8). We then evaluate both the objective function $f$ and the constraint functions $c_k$ at the proposed batch points $X_{\text{next}}$ (lines 9-10). We update the set of evaluated samples $\mathcal{S}$ (line 11) and we refit the surrogate models $\mathcal{M}$ and $\mathcal{C}_k$ over the full search space $\Omega$ using the updated sample set $\mathcal{S}$ (line 12).

We use the radius $R$ of the uniform distribution to dynamically adjust the scale of the search around $x_{\text{best}}$. In the distribution $\mathcal{I} \sim \text{Uniform}(x_{\text{best}}, R)$, we initialize $R = 1$. Given that the domain $\Omega$ is normalized to $[0, 1]^D$ in our implementation, the initial distribution of samples covers the entire domain, regardless of the exact position of $x_{\text{best}}$. Similarly to SCBO, two counters are maintained to track optimization progress (line 13): $n_s$, the number of successes (iterations where the best solution improves), and $n_f$, the number of failures (iterations without improvement). At each iteration, the radius is updated based on the following rules (lines 14-15): it is doubled if $n_s = \tau_s$ it is halved if $n_f = \tau_f$, it remains unchanged otherwise. After each update, both counters are reset $(n_s \leftarrow 0, n_f \leftarrow 0)$. The thresholds $\tau_s$ and $\tau_f$ are user-defined hyperparameters controlling the frequency of zoom-in/zoom-out behavior. A very small radius indicates stagnation or convergence. Optimization will be stopped or restarted when $R \leq \varepsilon$, with $\varepsilon$ chosen by the user.

## 5 Experiments

### 5.1 Experimental setup

We evaluate the performance of FuRBO against the following state-of-the-art methods: Scalable Constrained Bayesian Optimization (SCBO) by Eriksson and Poloczek [2021], constrained Expected Improvement (cEI) introduced by Schonlau et al. [1998], Constrained Optimization by Linear Approximation (COBYLA) from Powell [1994], constrained Covariance Matrix Adaptation Evolution Strategy (CMA-ES) by Hansen [2006], and a Random Search (for the URLs of the used implementations, see the References; our code is available on GitHub[2]). We compare these algorithms on the constrained black-box optimization benchmarking (BBOB-constrained) suite from the COCO package [Hansen et al., 2021]. The results of the constrained BBOB benchmark for FuRBO and SCBO are discussed in Sec. 5.2.1. The results comparing FuRBO to multiple baselines are discussed in Sec. 5.2.2. In Appendix F, we extend our comparison between FuRBO and SCBO to include the 30-dimensional Keane bump synthetic benchmark, as well as several physics-inspired problems with dimensionalities ranging from 3 to 60.

**Constrained BBOB**. We use the COCO/BBOB-constrained benchmark [Hansen et al., 2021], comprising 4,860 constrained black-box functions generated by combining 9 base functions, 6 constraint sets of increasing severity, 6 dimensions, and 15 instances [Dufossé and Atamna, 2022]. The functions are defined on a continuous search space and present different landscape characteristics (separable, ill-conditioned, multi-modal functions). For our evaluation, we use the full suite to compare FuRBO with its closest relative, SCBO. We consider 3 instances and 10 repetitions with different random seed per function-constraint combination in dimensions 2, 10, and 40.

For comparisons against all the mentioned baselines, we select three representative functions from the suite: Sphere (separable), Bent Cigar (ill-conditioned), and Rotated Rastrigin (multi-modal) in 10 dimensions, each with medium-complex constraint structures. The same experimental setup (initial design size 3D for the BO-based algorithms and initial sample 30D) is applied to all algorithms.

**Baselines Setup**. FuRBO and SCBO are evaluated on these functions, each repeated 10 times with different initial designs and random seeds. The initial design size is 3D, the batch size is 3D, and the total evaluation budget is 30D, where D is the problem dimension. FuRBO uses a TR defined from the top 10% of inspectors sampled around the current best solution. The sampling radius for the inspectors $R$ is initialized to 1, doubled when the success counter reaches $\tau_s = 2$, and halved when the failure counter reaches $\tau_f = 3$. Optimization restarts if $R$ reaches a minimum threshold $\varepsilon = 5 \times 10^{-8}$. CMA-ES and COBYLA are initialized using the default hyperparameter settings recommended by their respective implementations [Hansen et al., 2019, Virtanen et al., 2020]. For random sampling, we use a uniform distribution over the search space.

**Performance metrics**. We evaluate performance in terms of loss (simple regret), averaging over 30 runs with one standard error, and CPU time. Any feasible solution is preferred over infeasible ones, which are assigned the worst observed objective value for a specific problem setting across all compared methods [Hernández-Lobato et al., 2017]. CMA-ES and COBYLA are initialized from the best point from the initial sample set generated for the BO methods.

**Hardware and Runtime**. All experiments are conducted on an Intel i9-12900K 3.20GHz CPU. To provide an example of FuRBO runtime, the compute time for the constrained BBOB 10D functions spanned from 5.2 sec to 250 sec on CPU, depending on the complexity of the function landscape and the severity of the constraint. This gave a total of 33 h on CPU.

### 5.2 Results

**5.2.1 Constrained BBOB in 10D**. Figure 2 presents the loss (simple regret) convergence curves for FuRBO and SCBO across the full constrained BBOB suite in 10 dimensions. Both algorithms are

---

[2]https://github.com/paoloascia/FuRBO

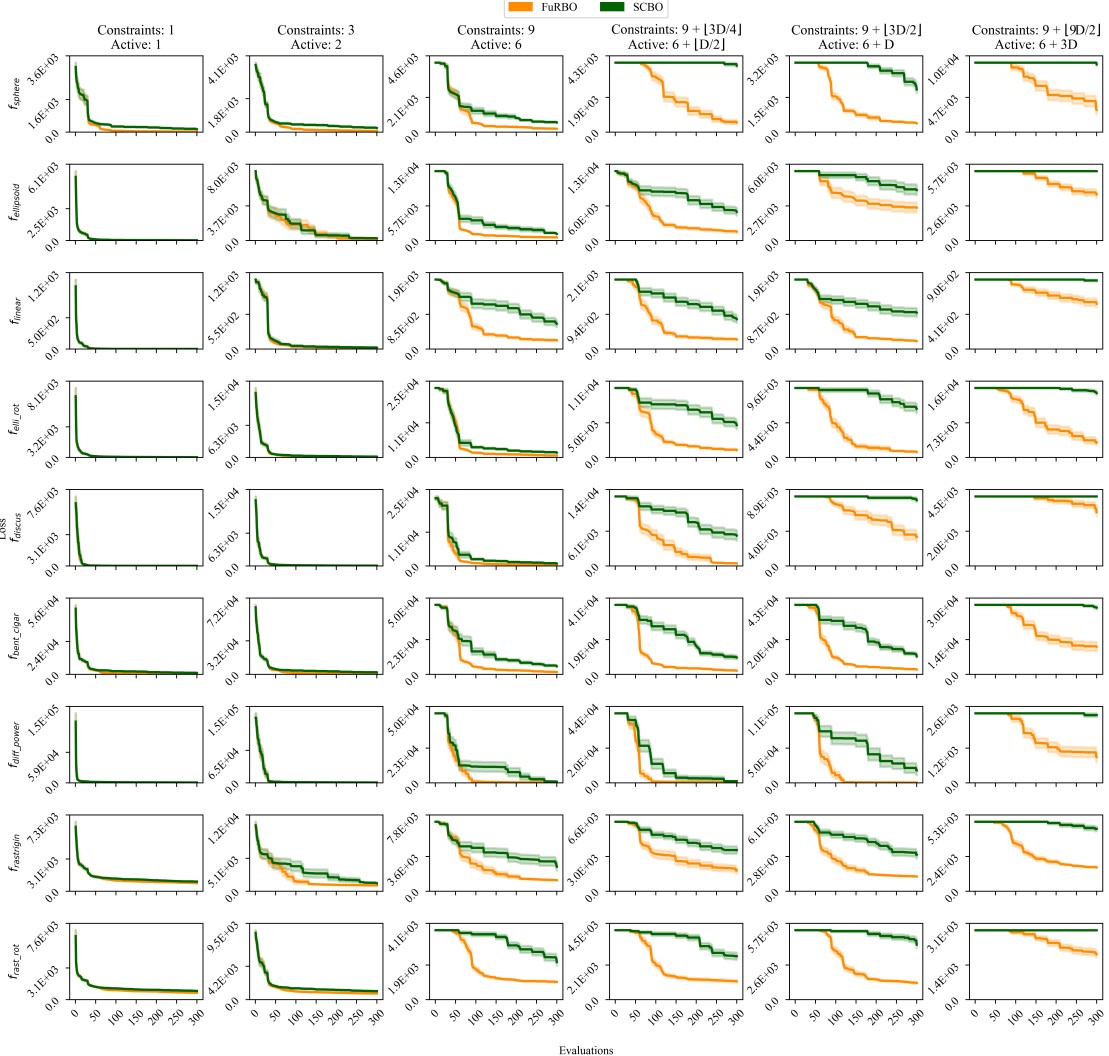

Figure 2: Loss convergence curve on the full constrained BBOB suite at 10D. Results are averaged across 3 instances with 10 repetitions each. The plot shows the mean loss with shaded areas indicating one standard error. FuRBO consistently outperforms SCBO on more severely constrained problems and performs comparably on easier ones.

run with a batch size $q = 3D$, and total evaluation budget of $30D$ to mimic real-world scenarios where function evaluations rely on very expensive procedures that can be run on parallel nodes (in Appendix E.4 we provide an ablation study on the batch size $q$).

Overall, FuRBO consistently outperforms SCBO on problems with a higher number of constraints and active constraints (rightmost columns), indicating its superior performance in severely constrained scenarios. Notably, for configurations with 17 or more constraints, FuRBO achieves faster convergence and lower final regret. For simpler problems (leftmost columns with 1–3 constraints), FuRBO performs comparably to SCBO, and in some cases the two methods are nearly indistinguishable in terms of convergence speed and final performance. The exact final performances of FuRBO and SCBO are reported in Table 2 in Appendix A, where we also assess statistical significance using the Wilcoxon rank-sum test. The analysis confirms that FuRBO achieves significantly better performance on the majority of the 10D problems. Similar figures to Figure 2, but for 2 and 40 dimensions are available in Appendix C. While FuRBO and SCBO perform similarly

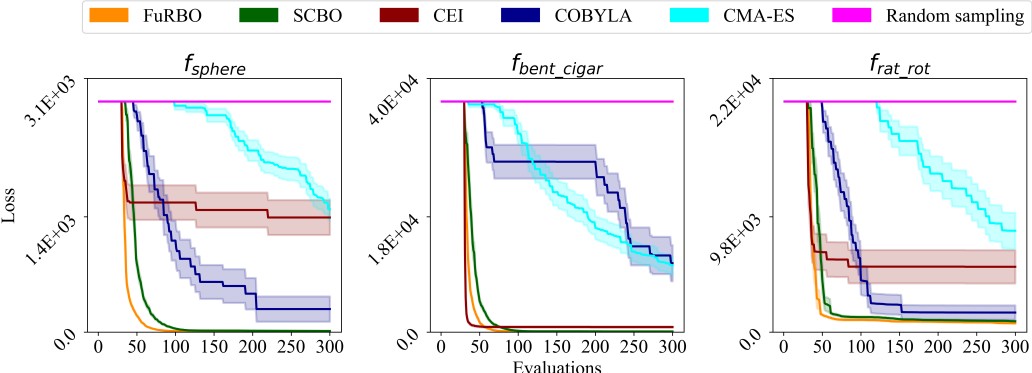

Figure 3: Convergence comparison of FuRBO against SCBO, CEI, COBYLA, CMA-ES, and random sampling on $f_{sphere}$, $f_{bent\_cigar}$, and $f_{rast\_rot}$ in 10D. Curves show the mean loss over 10 repetitions of the same instance, with shaded regions indicating one standard error.

in 2D, FuRBO shows clear superiority in 40D, where it succeeds in finding feasible solutions in cases where SCBO fails within the given evaluation budget. However, in the most strongly constrained scenarios, FuRBO also struggles to identify feasible regions, suggesting that the chosen hyperparameter settings may be not optimal for such cases. We will further investigate this.

The improvement observed in highly constrained cases highlights the effectiveness of FuRBO's feasibility-aware TR strategy. Unlike SCBO, FuRBO defines its TR based on the area predicted to be most promising, considering both feasibility and optimality predicted by the surrogate models over the entire domain, rather than relying solely on the best evaluated sample. This allows the TR to shift more freely across the domain and adapt its size dynamically: it contracts or expands according to the predicted distribution of high-quality, feasible regions. As a result, FuRBO is better able to zoom in on narrow feasible areas and escape local minima, offering faster convergence and more robust performance in complex, constrained landscapes.

**5.2.2 Comparison to SOTA baselines**. Figure 3 shows the convergence of FuRBO compared to SCBO, CEI, COBYLA, CMA-ES, and random sampling on three representative BBOB-constrained functions in 10 dimensions: Sphere, Bent Cigar, and Rotated Rastrigin. In order to have a comparable setup for all methods, we consider a batch size $q = 1$ for the BO methods (FuRBO, SCBO, and CEI), meaning that only one candidate solution is returned by the AF and evaluated at each iteration.

FuRBO consistently achieves the lowest final loss and fastest convergence across all cases, closely followed by SCBO. This highlights that the new definition of the TR introduced in FuRBO is more beneficial when multiple solutions, potentially spread within the TR, are returned at each iteration. On the ill-conditioned $f_{bent\_cigar}$ problem, CEI converges to low-loss regions faster than all baselines, however, FuRBO demonstrates greater exploitation capabilities by converging to a solution with a statistically significant lower value of the objective function. For the multimodal $f_{rast\_rot}$, FuRBO again outperforms the rest, directly followed by SCBO. CMA-ES and random sampling perform poorly across all functions, highlighting the benefit of using surrogate models in severely constrained and expensive settings. These results are confirmed in Table 4 in Appendix A.

## 6 Conclusions

We introduced FuRBO, a trust-region-based BO algorithm for severely constrained black-box problems. Building on the SCBO framework, it redefines the trust region using a feasibility-aware ranking of samples drawn uniformly from a ball centered around the current best point. This allows FuRBO to dynamically adapt the location and size of the trust region based on global

surrogate information about both the objective and constraint functions, accelerating convergence and improving exploitation of narrow feasible regions.

Our experiments on the BBOB-constrained benchmark suite demonstrate that FuRBO consistently outperforms or matches the performance of existing state-of-the-art methods, particularly in scenarios involving tight or complex constraints.

However, FuRBO also comes with a few limitations and directions for future work. First, the inspector-based ranking and trust region construction introduce additional model evaluations, making the algorithm computationally more expensive than other baselines, thus most appropriate when function evaluations are costly and dominate the runtime. Second, FuRBO still faces challenges in high-dimensional, heavily constrained problems, where it sometimes fails to find feasible regions. Third, our approach is still limited to a single axis-aligned trust region—a limitation we aim to overcome in future work by exploring more flexible and potentially multiple trust regions with adaptive orientations based on landscape characteristics. Finally, while we have evaluated our method on several benchmark problems inspired by physics and real-world engineering applications (e.g., spring design, pressure vessels, route planning), we will extend our validation to more complex scenarios that originally motivated this work—specifically, structural systems subject to crash constraints, modeled using the solution space approach [Zimmermann and De Weck, 2021].

**Acknowledgements**. The project leading to this application has received funding from the European Union's Horizon 2020 research and innovation program under the Marie Skłodowska-Curie grant agreement No 955393. The authors gratefully acknowledge the computational and data resources provided by the Leibniz Supercomputing Centre (www.lrz.de).

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

# A  Statistical evaluation

Here, we show the exact loss values reached by the algorithms at the end of the evaluation budget.

Table 1 presents the mean and standard error of the final objective values achieved by FuRBO and SCBO across constrained BBOB problems in 2D. Table 2 and Table 3 show similar results for dimensions 10 and 40, respectively. The results span increasing levels of constraint severity—ranging from 1 to $9 + \lfloor 9D/2 \rfloor$ constraints with varying numbers of active constraints and diverse problem landscapes (e.g., separable, ill-conditioned, multimodal). The best performance per setting is highlighted in bold, and statistical significance (via the Wilcoxon rank-sum test) is indicated in gray shading.

Table 1: Mean and standard error of the final performance across constrained BBOB problems in 2D. Best result is highlighted in bold. Statistical significance is assessed using the Wilcoxon rank-sum test. Results in gray indicate when one algorithm statistically outperforms the other.

| | Constraints: 1 (Active: 1) | | | | Constraints: 3 (Active: 2) | | | | Constraints: 9 (Active: 6) | | | |
|---|---|---|---|---|---|---|---|---|---|---|---|---|
| | FuRBO | | SCBO | | FuRBO | | SCBO | | FuRBO | | SCBO | |
| | Mean | S.E. | Mean | S.E. | Mean | S.E. | Mean | S.E. | Mean | S.E. | Mean | S.E. |
| $f_{sphere}$ | **0.210** | 0.040 | 0.527 | 0.079 | **2.218** | 0.231 | 4.000 | 0.561 | **4.181** | 0.721 | 6.794 | 0.987 |
| $f_{ellipsoid}$ | **0.210** | 0.038 | 0.249 | 0.054 | **8.843** | 1.067 | 16.295 | 2.378 | **19.783** | 2.363 | 49.156 | 7.773 |
| $f_{linear}$ | **0.018** | 0.003 | 0.040 | 0.006 | **1.615** | 0.255 | 2.711 | 0.397 | **2.748** | 0.261 | 4.518 | 0.413 |
| $f_{elli\_rot}$ | **1.213** | 0.242 | 1.723 | 0.339 | **11.792** | 2.008 | 18.158 | 3.125 | **16.756** | 1.427 | 50.838 | 4.230 |
| $f_{discus}$ | 0.645 | 0.169 | **0.550** | 0.086 | **6.643** | 0.688 | 13.904 | 0.971 | **20.132** | 4.756 | 46.491 | 8.410 |
| $f_{bent\_cigar}$ | 1.270 | 0.555 | **1.196** | 0.226 | **1.052** | 0.212 | 1.844 | 0.329 | **55.458** | 11.197 | 123.389 | 14.978 |
| $f_{diff\_power}$ | **2.828** | 1.185 | 3.306 | 1.054 | 10.584 | 3.024 | 7.576 | 1.116 | **14.498** | 2.651 | 16.235 | 2.274 |
| $f_{rastrigin}$ | **93.082** | 16.884 | 93.329 | 11.818 | **57.056** | 10.020 | 71.572 | 12.851 | **76.061** | 17.027 | 88.406 | 12.570 |
| $f_{rast\_rot}$ | **73.645** | 9.311 | 82.339 | 9.907 | **46.497** | 10.778 | 73.251 | 11.686 | 56.962 | 11.597 | **54.464** | 5.800 |
| | Constraints: $9 + \lfloor 3D/4 \rfloor$ (Active: $6 + \lfloor D/2 \rfloor$) | | | | Constraints: $9 + \lfloor 3D/2 \rfloor$ (Active: $6 + D$) | | | | Constraints: $9 + \lfloor 9D/2 \rfloor$ (Active: $6 + 3D$) | | | |
| | FuRBO | | SCBO | | FuRBO | | SCBO | | FuRBO | | SCBO | |
| | Mean | S.E. | Mean | S.E. | Mean | S.E. | Mean | S.E. | Mean | S.E. | Mean | S.E. |
| $f_{sphere}$ | **5.744** | 1.631 | 14.549 | 2.023 | **6.572** | 1.152 | 14.143 | 2.994 | **6.274** | 0.804 | 7.488 | 0.928 |
| $f_{ellipsoid}$ | **12.433** | 1.263 | 22.620 | 2.009 | **24.394** | 2.351 | 42.980 | 3.945 | **39.976** | 7.470 | 67.786 | 9.901 |
| $f_{linear}$ | **3.328** | 0.307 | 8.648 | 0.772 | **5.537** | 0.470 | 8.703 | 0.829 | **5.183** | 0.622 | 17.188 | 1.581 |
| $f_{elli\_rot}$ | **33.862** | 4.473 | 63.640 | 7.423 | **23.085** | 3.515 | 46.577 | 8.061 | **19.182** | 2.509 | 49.176 | 6.168 |
| $f_{discus}$ | **182.689** | 53.678 | 346.352 | 111.304 | **24.937** | 1.916 | 83.970 | 10.258 | **20.889** | 3.112 | 56.955 | 7.429 |
| $f_{bent\_cigar}$ | **60.131** | 5.471 | 94.093 | 16.005 | **89.175** | 18.078 | 104.077 | 10.872 | **94.550** | 28.501 | 427.606 | 341.950 |
| $f_{diff\_power}$ | 28.864 | 4.662 | **20.930** | 2.344 | **12.032** | 1.548 | 23.973 | 3.615 | **16.172** | 4.186 | 20.328 | 2.068 |
| $f_{rastrigin}$ | 120.532 | 16.204 | **102.261** | 14.164 | **46.561** | 7.852 | 65.294 | 6.302 | **113.930** | 16.120 | 194.695 | 12.615 |
| $f_{rast\_rot}$ | 111.154 | 12.988 | **92.853** | 11.593 | **82.750** | 16.000 | 136.416 | 12.411 | **52.617** | 10.019 | 81.268 | 8.328 |

We observe that, nearly on all problems and across all levels of constraint severity, FuRBO outperforms or matches SCBO, often by a statistically significant margin. Notably:

- The superiority of FuRBO becomes evident as the dimensionality of the problem and the severity of the constraint increases.

- In dimension 2, despite the final loss values achieved by SCBO are sometimes better under mild constraints, the performance is nevere statistically better than the one of FuRBO.

- In dimension 2 and 10, under the most stringent settings FuRBO demonstrates robust better performance, significantly outperforming SCBO, where SCBO also exhibits a larger variance.

- In dimension 40, both FuRBO and SCBO fail to converge effectively for highly severe constraint settings ($9 + \lfloor 3D/2 \rfloor$ and $9 + \lfloor 9D/2 \rfloor$ constraints). However, FuRBO manages to find feasible solutions for $9 + \lfloor 3D/4 \rfloor$ constraints on nearly all benchmarks, while SCBO systematically fails.

These findings highlight FuRBO's superior adaptability in severely constrained scenarios, thanks to its feasibility-aware TR strategy. Also, the consistent performance across diverse problem types and constraint severities suggest that the algorithm generalizes well, effectively balancing global exploration with local refinement.

Table 2: Mean and standard error of the final performance across constrained BBOB problems in 10D. Best result is highlighted in bold. Statistical significance is assessed using the Wilcoxon rank-sum test. Results in gray indicate when one algorithm statistically outperforms the other. If a feasible solution is not returned, the results are marked as not available (n/a).

| | Constraints: 1 (Active: 1) | | | | Constraints: 3 (Active: 2) | | | | Constraints: 9 (Active: 6) | | | |
|---|---|---|---|---|---|---|---|---|---|---|---|---|
| | FuRBO | | SCBO | | FuRBO | | SCBO | | FuRBO | | SCBO | |
| | Mean | S.E. | Mean | S.E. | Mean | S.E. | Mean | S.E. | Mean | S.E. | Mean | S.E. |
| $f_{sphere}$ | **21.964** | 1.388 | 138.804 | 10.956 | **58.790** | 5.696 | 219.152 | 15.692 | **198.514** | 18.634 | 585.673 | 39.943 |
| $f_{ellipsoid}$ | **5.831** | 0.481 | 19.519 | 1.359 | **138.762** | 20.431 | 231.414 | 24.137 | **512.837** | 34.019 | 1.083e+03 | 61.844 |
| $f_{linear}$ | 0.122 | 0.022 | **0.087** | 0.017 | **12.195** | 1.571 | 21.249 | 1.870 | **212.422** | 16.858 | 618.493 | 86.203 |
| $f_{elli\_rot}$ | **13.956** | 1.076 | 22.265 | 1.618 | **105.630** | 13.958 | 139.813 | 14.482 | **611.267** | 71.576 | 1.583e+03 | 142.554 |
| $f_{discus}$ | **0.523** | 0.098 | 0.638 | 0.087 | **18.162** | 2.627 | 31.136 | 3.379 | **288.711** | 17.185 | 765.207 | 71.978 |
| $f_{bent\_cigar}$ | **156.625** | 11.214 | 1.011e+03 | 96.125 | **385.712** | 53.371 | 1.810e+03 | 142.890 | **1.453e+03** | 140.637 | 5.233e+03 | 480.111 |
| $f_{diff\_power}$ | **113.271** | 10.430 | 456.299 | 30.881 | **121.649** | 9.565 | 470.231 | 29.952 | **196.147** | 13.799 | 693.546 | 72.736 |
| $f_{rastrigin}$ | **769.783** | 18.940 | 913.273 | 19.361 | **887.369** | 32.102 | 1.209e+03 | 62.581 | **1.128e+03** | 40.189 | 2.517e+03 | 379.334 |
| $f_{rast\_rot}$ | **674.252** | 20.913 | 858.090 | 15.381 | **767.100** | 23.015 | 1.056e+03 | 33.444 | **958.011** | 34.706 | 2.033e+03 | 181.885 |

| | Constraints: 9 + ⌊3D/4⌋ (Active: 6 + ⌊D/2⌋) | | | | Constraints: 9 + ⌊3D/2⌋ (Active: 6 + D) | | | | Constraints: 9 + ⌊9D/2⌋ (Active: 6 + 3D) | | | |
|---|---|---|---|---|---|---|---|---|---|---|---|---|
| | FuRBO | | SCBO | | FuRBO | | SCBO | | FuRBO | | SCBO | |
| | Mean | S.E. | Mean | S.E. | Mean | S.E. | Mean | S.E. | Mean | S.E. | Mean | S.E. |
| $f_{sphere}$ | **543.153** | 116.179 | 3.729e+03 | 114.165 | **369.019** | 15.963 | 1.805e+03 | 160.573 | **2.976e+03** | 643.900 | 9.106e+03 | 215.199 |
| $f_{ellipsoid}$ | **1.534e+03** | 160.652 | 4.952e+03 | 630.699 | **2.584e+03** | 395.692 | 3.967e+03 | 400.501 | **3.446e+03** | 250.820 | n/a | n/a |
| $f_{linear}$ | **258.470** | 29.091 | 809.796 | 103.165 | **196.186** | 9.024 | 909.622 | 108.059 | **535.789** | 44.294 | 811.818 | 10.493 |
| $f_{elli\_rot}$ | **1.056e+03** | 96.825 | 4.644e+03 | 601.538 | **680.446** | 45.009 | 6.133e+03 | 566.422 | **3.111e+03** | 587.884 | 1.347e+04 | 403.178 |
| $f_{discus}$ | **424.307** | 31.046 | 5.358e+03 | 907.447 | **3.337e+03** | 567.569 | 7.617e+03 | 239.395 | **3.164e+03** | 243.911 | n/a | n/a |
| $f_{bent\_cigar}$ | **1.955e+03** | 164.335 | 8.968e+03 | 1.072e+03 | **2.753e+03** | 204.529 | 1.023e+04 | 618.958 | **1.092e+04** | 1.831e+03 | 2.640e+04 | 758.083 |
| $f_{diff\_power}$ | **282.976** | 22.572 | 885.045 | 86.379 | **310.649** | 15.017 | 1.783e+04 | 6.714e+03 | **875.926** | 164.867 | 2.308e+03 | 59.269 |
| $f_{rastrigin}$ | **1.797e+03** | 258.449 | 3.554e+03 | 321.989 | **1.171e+03** | 25.633 | 2.911e+03 | 295.358 | **1.650e+03** | 36.467 | 4.308e+03 | 164.089 |
| $f_{rast\_rot}$ | **1.086e+03** | 70.368 | 2.582e+03 | 210.997 | **1.245e+03** | 64.740 | 4.105e+03 | 252.273 | **1.840e+03** | 136.922 | n/a | n/a |

Table 3: Mean and standard error of the final performance across constrained BBOB problems in 40D. Best result is highlighted in bold. Statistical significance is assessed using the Wilcoxon rank-sum test. Results in gray indicate when one algorithm statistically outperforms the other. If a feasible solution is not returned, the results are marked as not available (n/a).

| | Constraints: 1 (Active: 1) | | | | Constraints: 3 (Active: 2) | | | | Constraints: 9 (Active: 6) | | | |
|---|---|---|---|---|---|---|---|---|---|---|---|---|
| | FuRBO | | SCBO | | FuRBO | | SCBO | | FuRBO | | SCBO | |
| | Mean | S.E. | Mean | S.E. | Mean | S.E. | Mean | S.E. | Mean | S.E. | Mean | S.E. |
| $f_{sphere}$ | **191.905** | 9.743 | 1.235e+03 | 74.848 | **891.213** | 56.888 | 2.195e+03 | 120.958 | **2.846e+03** | 187.348 | 4.947e+03 | 284.815 |
| $f_{ellipsoid}$ | **72.660** | 4.706 | 199.098 | 15.666 | **1.471e+03** | 251.414 | 1.594e+03 | 116.509 | **1.332e+03** | 135.523 | 4.121e+03 | 429.439 |
| $f_{linear}$ | 0.120 | 0.026 | **0.063** | 0.019 | **47.281** | 9.592 | 133.901 | 28.004 | **333.052** | 23.110 | 857.128 | 100.898 |
| $f_{elli\_rot}$ | **83.392** | 7.957 | 249.737 | 23.016 | **1.064e+03** | 67.244 | 1.602e+03 | 113.814 | **5.368e+03** | 424.582 | 1.046e+04 | 1.250e+03 |
| $f_{discus}$ | **0.181** | 0.046 | 0.284 | 0.043 | **43.144** | 6.356 | 56.058 | 9.523 | **265.416** | 35.171 | 666.984 | 77.161 |
| $f_{bent\_cigar}$ | **1.770e+03** | 112.493 | 1.447e+04 | 720.873 | **1.704e+04** | 1.369e+03 | 3.033e+04 | 1.906e+03 | **5.530e+04** | 9.294e+03 | 1.130e+05 | 1.100e+04 |
| $f_{diff\_power}$ | **1.510e+03** | 80.505 | 2.793e+03 | 133.038 | **2.417e+03** | 200.116 | 6.807e+03 | 500.896 | **5.250e+03** | 759.436 | 2.413e+04 | 3.647e+03 |
| $f_{rastrigin}$ | **3.804e+03** | 146.398 | 4.959e+03 | 101.575 | **4.743e+03** | 138.682 | 6.602e+03 | 167.442 | **5.936e+03** | 197.260 | 8.826e+03 | 181.602 |
| $f_{rast\_rot}$ | **4.318e+03** | 99.437 | 5.109e+03 | 161.808 | **5.227e+03** | 174.590 | 6.311e+03 | 152.997 | **8.753e+03** | 447.437 | 1.484e+04 | 814.410 |

| | Constraints: 9 + ⌊3D/4⌋ (Active: 6 + ⌊D/2⌋) | | | | Constraints: 9 + ⌊3D/2⌋ (Active: 6 + D) | | | | Constraints: 9 + ⌊9D/2⌋ (Active: 6 + 3D) | | | |
|---|---|---|---|---|---|---|---|---|---|---|---|---|
| | FuRBO | | SCBO | | FuRBO | | SCBO | | FuRBO | | SCBO | |
| | Mean | S.E. | Mean | S.E. | Mean | S.E. | Mean | S.E. | Mean | S.E. | Mean | S.E. |
| $f_{sphere}$ | **6.150e+03** | 539.698 | n/a | n/a | **5.735e+03** | 238.843 | n/a | n/a | n/a | n/a | n/a | n/a |
| $f_{ellipsoid}$ | **1.307e+04** | 1.064e+03 | n/a | n/a | n/a | n/a | n/a | n/a | n/a | n/a | n/a | n/a |
| $f_{linear}$ | **2.413e+03** | 263.602 | n/a | n/a | n/a | n/a | n/a | n/a | n/a | n/a | n/a | n/a |
| $f_{elli\_rot}$ | **1.837e+04** | 1.347e+03 | n/a | n/a | n/a | n/a | n/a | n/a | n/a | n/a | n/a | n/a |
| $f_{discus}$ | **1.381e+04** | 524.696 | n/a | n/a | n/a | n/a | n/a | n/a | n/a | n/a | n/a | n/a |
| $f_{bent\_cigar}$ | **7.508e+04** | 5.295e+03 | n/a | n/a | n/a | n/a | n/a | n/a | n/a | n/a | n/a | n/a |
| $f_{diff\_power}$ | **8.737e+03** | 2.276e+03 | n/a | n/a | n/a | n/a | n/a | n/a | n/a | n/a | n/a | n/a |
| $f_{rastrigin}$ | n/a | n/a | n/a | n/a | n/a | n/a | n/a | n/a | n/a | n/a | n/a | n/a |
| $f_{rast\_rot}$ | **1.191e+04** | 821.514 | n/a | n/a | n/a | n/a | n/a | n/a | n/a | n/a | n/a | n/a |

Table 4 reports the final performance of FuRBO and four other SOTA baselines: SCBO, CEI, COBYLA, and CMA-ES, on three representative benchmark functions in 10D: $f_{\mathrm{sphere}}$, $f_{\mathrm{bent\_cigar}}$, and $f_{\mathrm{rast\_rot}}$. We omit random sampling because it never found a feasible solution in our comparison. These chosen functions represent diverse landscape characteristics, from simple convex and separable functions, to ill-conditioned and multimodal. We highlight the best-performing algorithm in bold, with statistical significance (via a pairwise Wilcoxon rank-sum test) indicated by gray shading. This means that we highlight in gray a method only if it statistically outperformed all the others in a problem setting.

On $f_{\text{sphere}}$, a smooth, convex, and separable problem, FuRBO clearly outperforms SCBO and dramatically outpaces CEI, COBYLA, and CMA-ES. On the ill-conditioned $f_{\text{bent\_cigar}}$ function, FuRBO again leads with a significantly lower final objective value than all baselines, highlighting its capacity to exploit narrow feasible valleys. For the multimodal $f_{\text{rast\_rot}}$, FuRBO maintains its lead, demonstrating superior exploration of complex feasible regions and local minima. However, the final loss value significantly increases compared to the two simpler functions. These results further highlight FuRBO's strength, not only over its direct competitor SCBO, but also against well-established baselines fro constrained optimization from the literature.

Table 4: Mean and standard error of the final performance on the class representative functions in 10D. The best result for each function is highlighted in bold. Statistical significance is assessed using the Wilcoxon rank-sum test. Results in gray indicate when one algorithm statistically outperforms all the others.

|  | $f_{sphere}$ | | $f_{bent\_cigar}$ | | $f_{rast\_rot}$ | |
|---|---|---|---|---|---|---|
|  | Mean | S.E. | Mean | S.E. | Mean | S.E. |
| FuRBO | **7.286** | 0.580 | **68.944** | 5.132 | **760.095** | 38.507 |
| SCBO | 14.237 | 0.696 | 107.133 | 5.568 | 925.548 | 47.247 |
| CEI | 829.146 | 87.943 | 798.386 | 96.740 | 2093.581 | 270.394 |
| COBYLA | 126.358 | 86.315 | 5787.570 | 1536.906 | 1156.212 | 126.745 |
| CMA-ES | 1211.966 | 116.424 | 8874.131 | 1200.243 | 3331.974 | 350.704 |

## B  Scalability of FuRBO

Figure 4 illustrates the mean loss convergence of FuRBO and SCBO on three constrained BBOB functions: $f_{\text{sphere}}$, $f_{\text{bent\_cigar}}$, and $f_{\text{rast\_rot}}$, evaluated at a mid-severe constraint level ($9 + \lfloor 3D/4 \rfloor$ constraints, out of which $6 + \lfloor D/2 \rfloor$ are active) in dimensions 2, 10, and 40. The results are averaged across 3 problem instances and 10 repetitions each, with shaded areas representing one standard error.

FuRBO demonstrates clear scalability advantages as dimensionality increases. In 2D, FuRBO and SCBO perform similarly, with FuRBO showing slightly faster convergence in the early evaluations. In 10D and 40D, FuRBO consistently outperforms SCBO, achieving lower final losses and faster convergence.

We note that, particularly in higher dimensions, the convergence curves exhibit a step-wise pattern. This behavior is due to the chosen batch size of $q = 3D$ and the way solutions within each batch are automatically ordered by objective value (from worst to best) by the `BoTorch` package, from which we took the SCBO implementation and on which we also base our own. As a result, early points in each batch tend to be of lower quality and are often outperformed by previously found solutions, leading to flat segments in the convergence curves. Instead, the last solutions returned in the batch, which are of higher quality, typically bring an improvement.

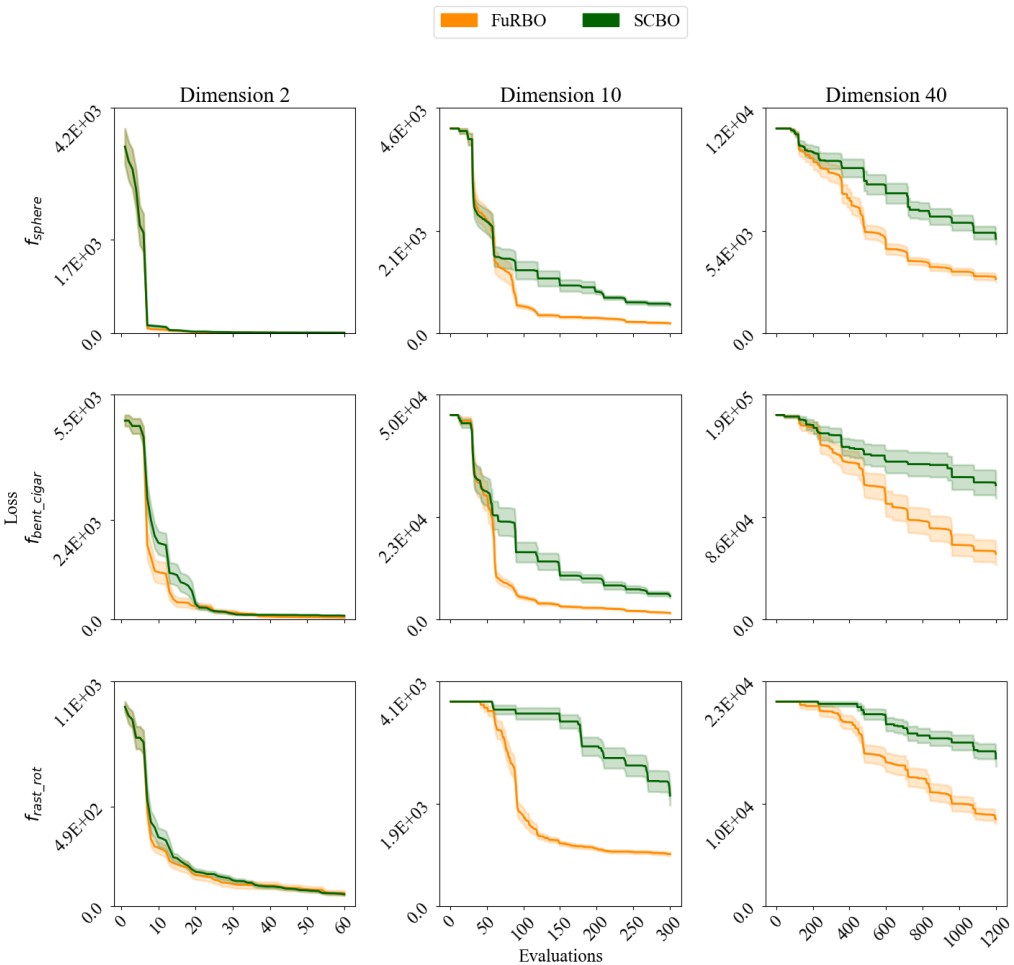

Figure 4: Loss convergence curve on $f_{\text{sphere}}$, $f_{\text{bent\_cigar}}$, and $f_{\text{rast\_rot}}$ constrained BBOB functions at 10D and mid-severe constraint level, for varying dimensionality of the problems. Results are averaged across 3 instances with 10 repetitions each. The plot shows the mean loss with shaded areas indicating one standard error. FuRBO clearly outperforms SCBO in 10D and 40D, while it shows comparable performance to SCBO in 2D, with slightly faster convergence.

## C  Full results on Constrained BBOB

In this section, we show the comparison between FuRBO and SCBO on the full constrained BBOB test suite for dimension 10.

In 2D (Figure 5), both FuRBO and SCBO perform similarly across most functions and constraint levels.

In 40D (Figure 6), the difference becomes more pronounced. FuRBO outperforms SCBO under moderate constraint counts. This is particularly evident on the 9 constraints (6 active) column. As the constraint severity increases, SCBO stops finding any feasible solutions, as indicated by the constant convergence trend, while FuRBO continues to make progress. In more extreme cases ($9 + \lfloor 3D/2 \rfloor$ or $9 + \lfloor 9D/2 \rfloor$ constraints), both methods struggle to find feasible improvements within the available evaluation budget.

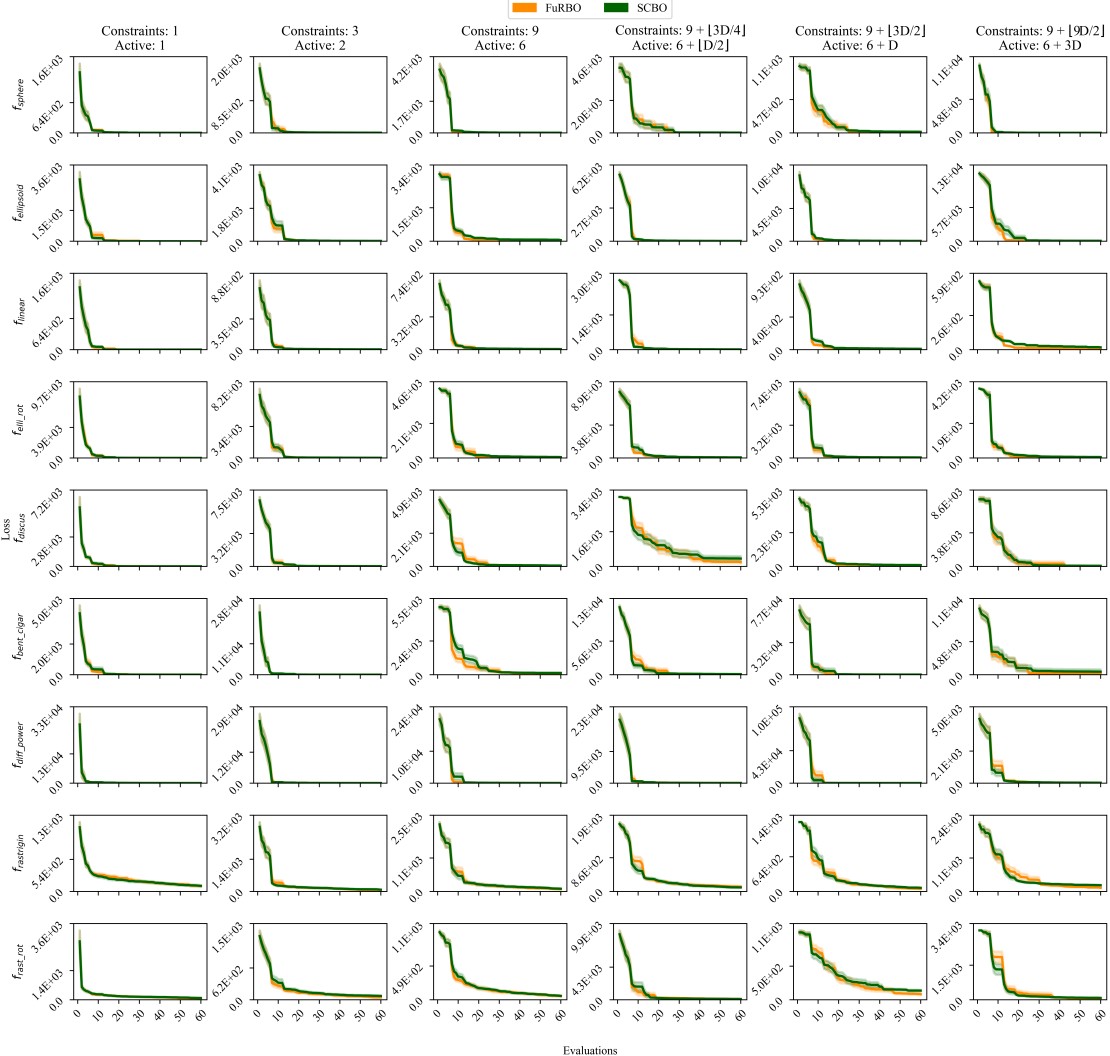

Figure 5: Loss convergence curve on the full constrained BBOB suite at 2D. Results are averaged across 3 instances with 10 repetitions each. The plot shows the mean loss with shaded areas indicating one standard error. FuRBO and SCBO have comparable convergence trends.

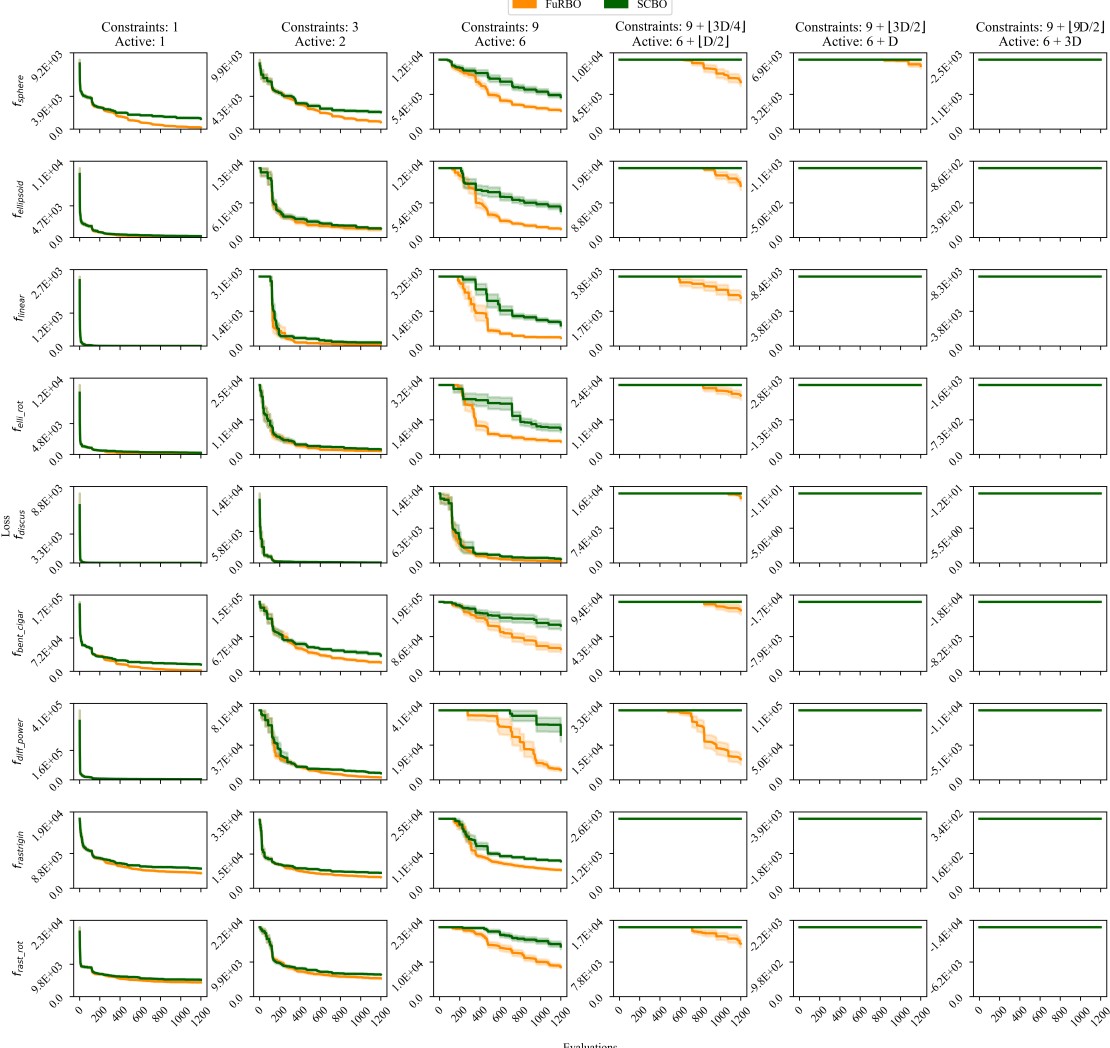

Figure 6: Loss convergence curve on the full constrained BBOB suite at 40D. Results are averaged across 3 instances with 10 repetitions each. The plot shows the mean loss with shaded areas indicating one standard error. FuRBO is on par or outperforms SCBO for mild and medium-severe constraints, while both methods fail in finding feasible solutions for strongly constrained scenarios.

# D CPU time

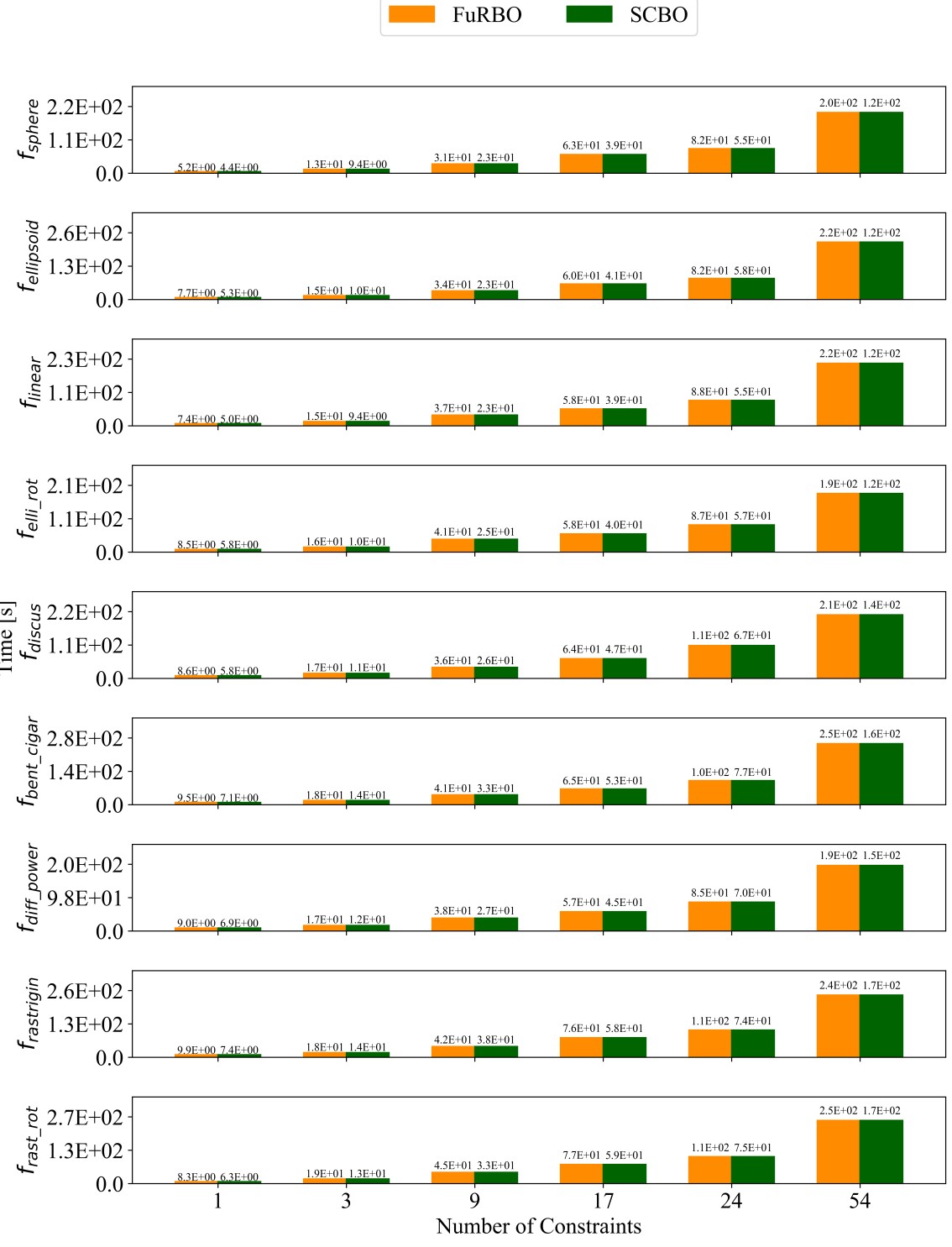

Figure 7: Average CPU time (in seconds) of FuRBO and SCBO across 10D all the constrained BBOB functions under increasing numbers of constraints. The two methods have comparable runtime, which increases with constraint severity.

Figure 7 presents the average CPU time required by FuRBO and SCBO in 10D, as the number of constraints and problem complexity increase. The batch size set for both algorithms is $q = 3D$.

Across all test functions, both methods show a similar scaling trend: runtime increases with the number of constraints, as expected due to the additional computational burden of constraint modeling and feasibility checking. FuRBO's computational cost presents only marginal increases compared to SCBO's, as it performs additional calls to the objective and constraint approximation models for the definition of the TR.

Figure 8 compares FuRBO and SCBO with other established baselines (CEI, COBYLA, and CMA-ES) on three representative functions in 10D and medium-high constraint severity ($9 + \lfloor 3D/4 \rfloor$ constraints, $6 + \lfloor D/2 \rfloor$ active). As expected, the surrogate-based methods (CEI, SCBO, and FuRBO) are by far the most expensive. COBYLA and CMA-ES are significantly faster, but at the cost of substantially poorer optimization performance (as shown in Figure 3 and Table 4).

Please note that the runtime tracked for FuRBO and SCBO for these settings seem in contrast with the one shown in Figure 8, for the same dimensionality. This is motivated by the fact that here, for comparison purposes, we are running all the methods (except CMA-ES, due to its inherently parallel design) in serial mode.

In summary, FuRBO offers a favorable balance between performance and runtime, finding better solutions than baselines like SCBO and CEI, while keeping computational overhead well within practical limits.

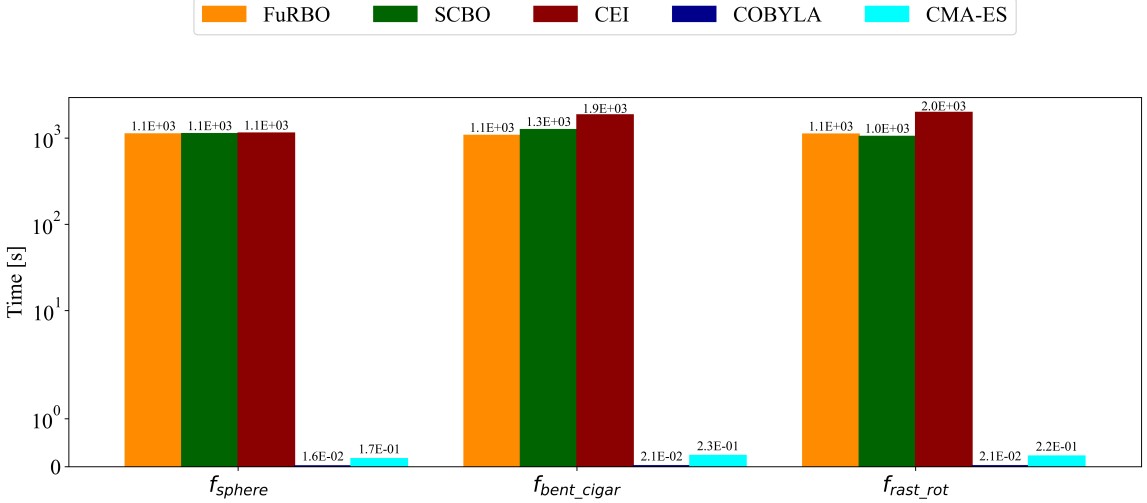

Figure 8: Average CPU time (log scale) of FuRBO against other four baselines (SCBO, CEI, COBYLA, and CMA-ES) on three representative 10D functions: $f_{\text{sphere}}$, $f_{\text{bent\_cigar}}$, and $f_{\text{rast\_rot}}$. FuRBO and SCBO have similar runtimes, while CEI is slightly more expensive. COBYLA and CMA-ES are much faster but at the cost of reduced solution quality (see Table 4).

## E  Ablation studies

In this section, we provide ablation studies on four influential hyperparameters of FuRBO. The size of the initial sample, the initial radius $R$ of the ball containing the uniformly distributed population of inspectors, the percentage of investigators used to define the trust region, and the batch size $q$. All the studies in this section are performed on the $f_{\text{bent\_cigar}}$ function with 24 constraints in dimension 10. We do not present a study on the effect of the number of investigators, as it did not show any noticeable impact in our experiments. We also experimented with using the total constraint violation, as done in SCBO [Eriksson et al., 2019], but observed no notable difference in

performance compared to our current ranking based on maximum normalized constraint violation. The corresponding plots are available in the GitHub repository.

## E.1  Initial sample size

Figure 9 presents a study on the impact of the size of the initial sample set, also referred to as Design of Experiments (DoE), on the optimization performance of FuRBO. We compare four DoE sizes proportional to the problem dimensionality, specifically, 1D, 3D, 5D, and 10D initial points.

The results show that larger DoE sizes (e.g., 10D) tend to delay early convergence, as more evaluations are used upfront for model initialization before optimization begins. In contrast, smaller DoE sizes (1D–5D) enable comparable (and faster compared to the 10D size) progress by allowing the iterative procedure to start earlier, without noticeably compromising the accuracy of the approximation models of the objective and constraints, and thus preserving the effectiveness of the landscape-aware mechanism for TR definition.

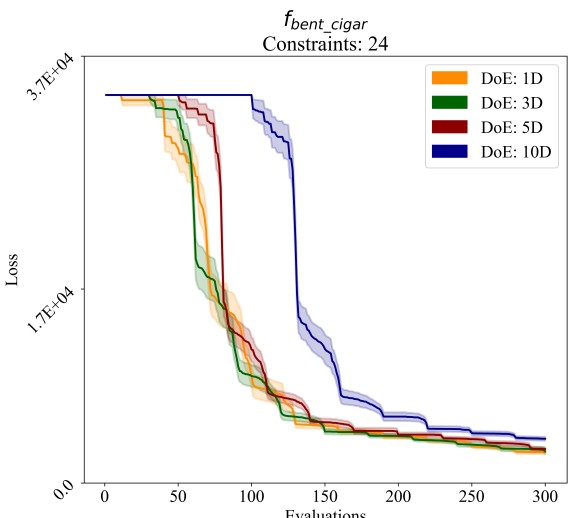

Figure 9: Study on the impact of the initial sample size on FuRBO in dimension 10 on the $f_{bent\_cigar}$ BBOB function with 24 constraints. Initial sample sizes equal to 1D, 3D, 5D and 10D are compared. Experiments run on $f_{bent\_cigar}$ BBOB function with 24 constraints.

## E.2  Sampling radius

Figure 10 shows the effects of changing the initial value of the sampling radius $R$ for the population of inspectors. We compare the performance of five different starting radii, $R = \{1.0, 0.5, 0.2, 0.1, 0.05\}$, on the constrained $f_{bent\_cigar}$ function (with 24 constraints).

The results show that, when FuRBO is initialized with a radius smaller than 0.5, the algorithm's performance deteriorates due to overlocalization of the search from the very first iterations, limiting its ability to search for feasible solutions and explore the landscape before convergence. No difference in performance was observed for $R = 1.0$ and $R = 0.5$ as both these choices allowed for a full coverage of the search space.

## E.3  Inspector percentage

Figure 11 investigates the effect of the percentage $P_\%$ of inspectors selected to define the TR in FuRBO. Specifically, we compare the performance on the constrained $f_{bent\_cigar}$ function (with 24 constraints) using selection rates of 1%, 5%, 10%, and 20%.

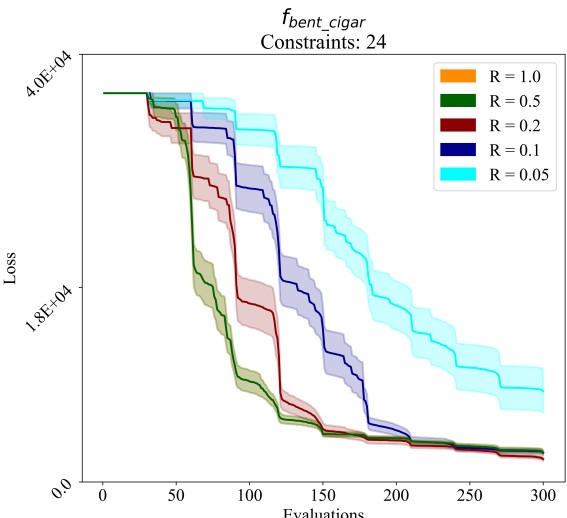

Figure 10: Study on the impact of the initial radius of the uniform distribution for the inspectors. The following initial radii are compared on the $f_{\text{bent\_cigar}}$ BBOB function with 24 constraints: $R = \{1.0, 0.5, 0.2, 0.1, 0.05\}$.

The results show that the algorithm's performance deteriorates as this percentage increases. However, we believe that these results highly depend on the landscape of the problem at hand. Very small percentages (e.g., 1%) enable faster initial progress by quickly narrowing down to promising regions, but they may limit the algorithm's ability to explore diverse feasible areas. Conversely, larger values (e.g., 20%) lead to slower progress, as the TR is influenced by a broader sample set, which may dilute the effect of high-quality candidates. Given this sensitivity, it would be beneficial to perform a dedicated hyperparameter optimization for this selection percentage to adapt it to the problem at hand.

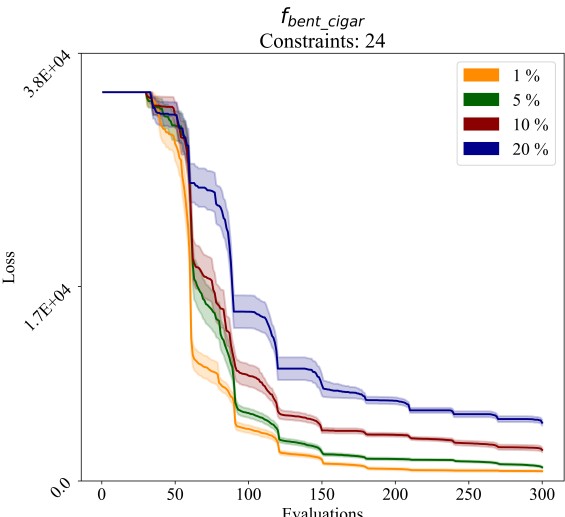

Figure 11: Study on the impact of the percentage $P_\%$ of inspectors selected to define the position and extension of the TR. The following percentages are compared: 1%, 5%, 10%, and 20%. Experiments run on $f_{\text{bent\_cigar}}$ BBOB function with 24 constraints.

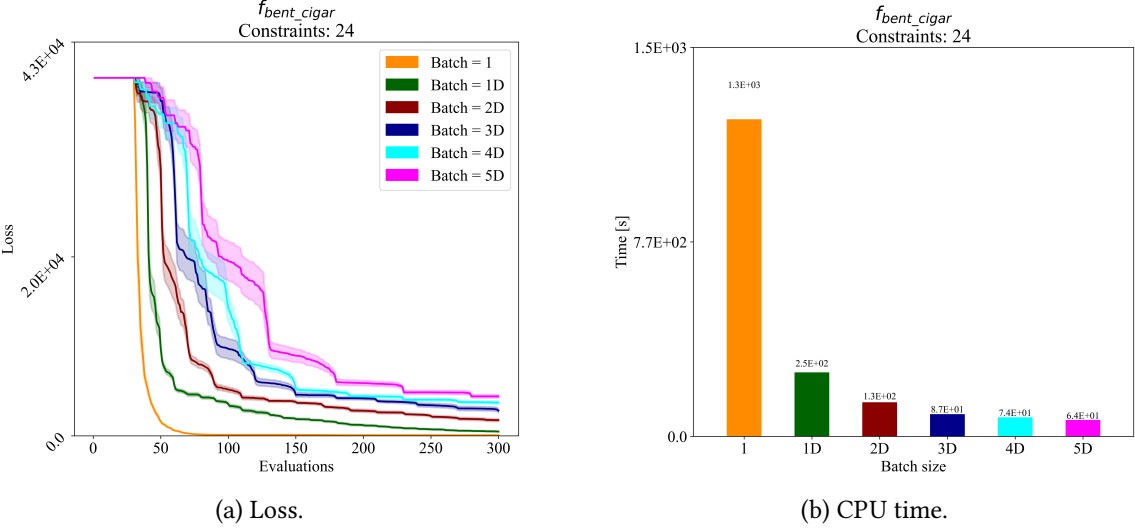

(a) Loss.  (b) CPU time.

Figure 12: Study on the impact of the batch size $q$ on the performance of FuRBO. The following configurations are compared: 1 (sequential), 1D, 2D, 3D, 4D, and 5D samples per batch. Experiments run on $f_{\text{bent\_cigar}}$ BBOB function with 24 constraints.

### E.4  Batch size

Figure 12 examines the impact of batch size $q$ on FuRBO's performance and computational cost. We compare six batch configurations: sequential (1 sample per iteration), and parallel batches of size 1D, 2D, 3D, 4D, and 5D.

Figure 12a shows that larger batch sizes generally lead to slower convergence as many samples are evaluated per iteration without the model being updated. The sequential setting (batch size = 1) achieves the fastest convergence in terms of evaluations, but as seen in Figure 12b, it incurs the highest computational cost due to frequent model updates and definitions of the TR.

Conversely, increasing the batch size significantly reduces CPU time, with a 5D batch being about 20 times faster than the sequential setup, while maintaining competitive final performance. Thus, intermediate batch sizes from 2D to 4D seem to provide a favorable trade-off, offering both efficiency and robust convergence behavior. FuRBO is therefore well-suited for optimization problems that require parallel evaluations of the objective and constraint functions, particularly when wall-clock time is a critical factor and parallel computing nodes are available.

## F  Results on other benchmarks

In this section, we present additional comparisons between FuRBO and SCBO. We include the minimization of the 30D Keane Bump function under two constraints, as well as several physics-inspired benchmark problems spanning dimensions from 3 to 60. For all problems, we independently ran both algorithms and report the results obtained from our own experiments.

### F.1  Keane bump function (30D)

We consider the Keane function [Keane, 1994] in 30 dimensions under two constraints over the domain $[0, 10]^D$:

$$\min \ f(x) = - \left| \frac{\sum_{i=1}^{D} \cos^4(x_i) - 2 \prod_{i=1}^{D} \cos^2(x_i)}{\sqrt{\sum_{i=1}^{D} i x_i^2}} \right|$$

$$\text{s.t.} \ c_1(x) = 0.75 - \prod_{i=1}^{D} x_i \leq 0,$$

$$c_2(x) = \sum_{i=1}^{D} x_i - 7.5d \leq 0. \tag{2}$$

The algorithms are evaluated with an initial DoE of 3D, a batch size of 3D, and a total budget of 30D. More specifically, FuRBO starts with an initial radius of 1.0 for the inspector population, and defines the TR by considering the top 10% of the investigators.

The results in Figure 13 indicate that FuRBO slightly outperforms SCBO on the 30D Keane Bump function, particularly in the later stages of the optimization. This suggests that FuRBO's trust region mechanism can offer benefits in high-dimensional settings. However, since the problem is only mildly constrained, the performance gap remains modest.

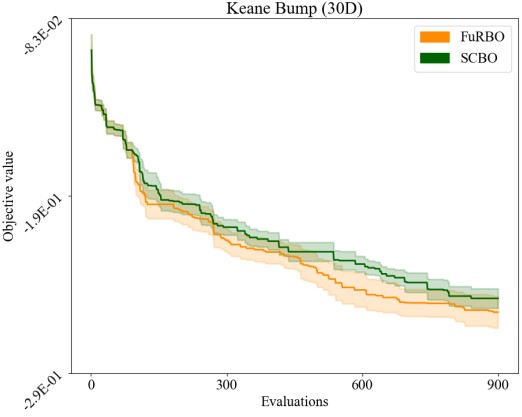

Figure 13: Convergence of FuRBO and SCBO on the 30D Keane Bump function under two constraints in the domain $[0, 10]^D$. Results are averaged over 10 independent runs for each algorithm. The plot shows the mean objective value, with shaded areas representing one standard error.

## F.2 Spring volume minimization (3D)

The first physics-inspired benchmark we evaluate is the minimization of the volume of a spring under four mechanical constraints [Lemonge et al., 2010]. The spring is described by three design parameters: the number of active coils of the spring $N \in [2, 15]$, the winding diameter $D_w \in [0.25, 1.3]$, and the wire diameter $d_w \in [0.05, 2]$. We compare the performance of FuRBO and SCBO with an initial DoE of 3D, a batch size of 3D, and a total budget of 30D.

Figure ??springfig springs the convergence behavior of FuRBO and SCBO on the 3D Spring design problem. Both methods steadily reduce the volume as the number of evaluations increases. While SCBO converges faster in the early stages and maintains a performance advantage throughout most of the optimization, the gap between the two methods is small. These results suggest that

FuRBO does not offer a significant benefit over SCBO in low-dimensional, lightly constrained problems, where the added complexity of its feasibility-driven trust region mechanism may be less impactful.

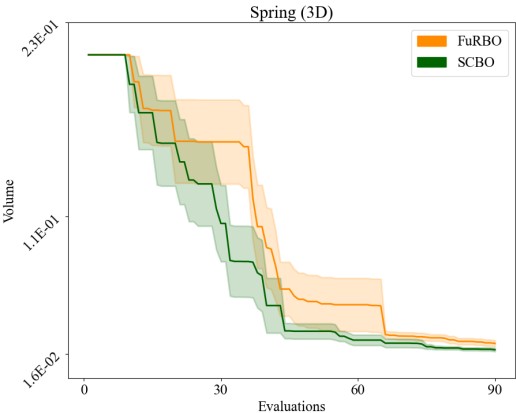

Figure 14: Convergence of FuRBO and SCBO on the 3D Spring design problem with three design variables, under four constraints. Results are averaged over 10 independent runs for each algorithm. The plot shows the mean objective value (volume), with shaded areas representing one standard error.

### F.3 Welded beam cost minimization (4D)

The second physics-inspired problem is the minimization of the welding costs for a beam under five mechanical constraints [Lemonge et al., 2010]. The design space is described by the length $l \in [0.1, 10.0]$ and the height $h \in [0.125, 10.0]$ of the welding, and the thickness $t \in [0.1, 10.0]$ and the width $b \in [0.1, 10.0]$ of the beam. Both FuRBO and SCBO are initialized with an initial DoE 3D samples, a batch size of 3D, and a total budget of 30D.

Figure 15 shows minimal difference between SCBO and FuRBO, which, as in the previous benchmark, can be attributed to the problem's low dimensionality and weak constraint structure.

### F.4 Pressure vessel mass minimization (4D)

The third physics-inspired is the mass minimization of a pressure vessel [Lemonge et al., 2010]. The problem is described by four parameters ($T_s$, $T_h$, $R$, $L$) and constrained by four functions. In this case, the thickness of the walls of the pressure vessel $T_s$ and the thickness of the wall of the head of the vessel $T_h$ vary in the range $[0.0625, 5]$, in constant steps of 0.0625. Instead, the inner radius of the vessel $R$ and the length of the cylindrical component $L$ are defined in the $[10, 200]$ interval and continuous. We compare the performance between FuRBO and SCBO with an initial DoE size of 3D, a batch size of 3D, and a total budget of 30D. Because of the low dimensionality and the weak constraints, the two algorithms show very similar performance (Figure 16).

### F.5 Speed reducer volume minimization (7D)

The speed reducer volume minimization [Lemonge et al., 2010] problem presents 7 parameters and 11 constraints. The parameters are the gear face width $b \in [2.6, 3.6]$, the teeth module $m \in [0.7, 0.8]$, the number of teeth on the pinion $n \in [17, 28]$, the length of the first shaft between the supporting bearings $l_1 \in [7.3, 8.3]$, the length of the second shaft between the supporting bearings $l_2 \in [7.8, 8.3]$, the diameter of the first shaft $d_1 \in [2.9, 3.9]$, and the diameter of the second shaft $d_2 \in [2.9, 3.9]$.

Although this problem has a low dimensionality, it is highly constrained. As the results in Figure 17 show, FuRBO outperforms SCBO by a significant margin.

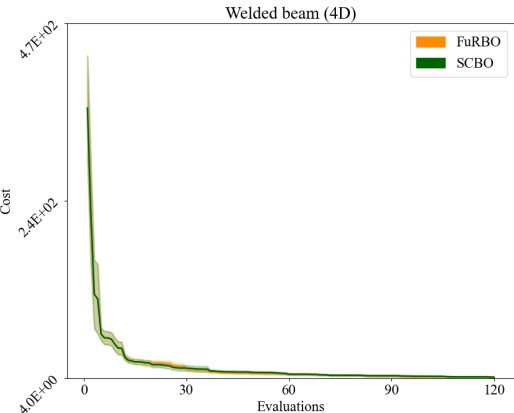

Figure 15: Convergence of FuRBO and SCBO on the 4D welded beam cost minimization problem, formulated with four design variables and five constraints. Results are averaged over 10 independent runs for each algorithm. The plot shows the mean objective value (cost), with shaded areas representing one standard error.

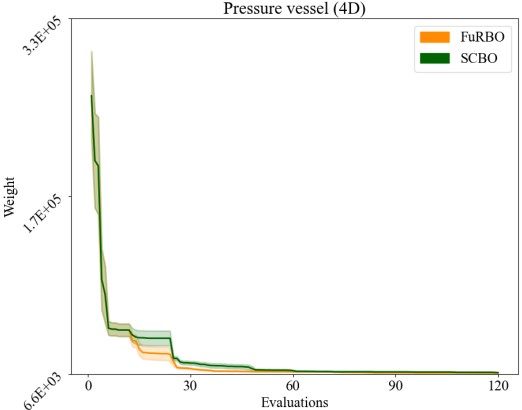

Figure 16: Convergence of FuRBO and SCBO on the pressure vessel design problem with four design parameters and four constraints. Results are averaged over 10 independent runs for each algorithm. The plot shows the mean objective value (weight), with shaded areas representing one standard error.

## F.6 Rover trajectory planning (60D)

The last benchmark on which we compare SCBO and FuRBO is the problem of planning the route of a rover over a terrain with different types of obstacles [Wang et al., 2018]. Similarly to Eriksson et al. [2019], we modify the original unconstrained trajectory optimization problem to include 15 hard constraints. The optimization problem is defined over a decision vector $x \in [0, 1]^{60}$, which encodes the trajectory $\gamma(x)$ of a rover navigating a grid environment. The environment contains 112 yellow obstacles, which the rover may cross at a cost, and 15 red obstacles, which must be avoided. A visualization of the problem definition is provided in Figure 18a, where the yellow squares represent the obstacles that the rover can overcome, while the red squares represent the obstacles to be avoided.

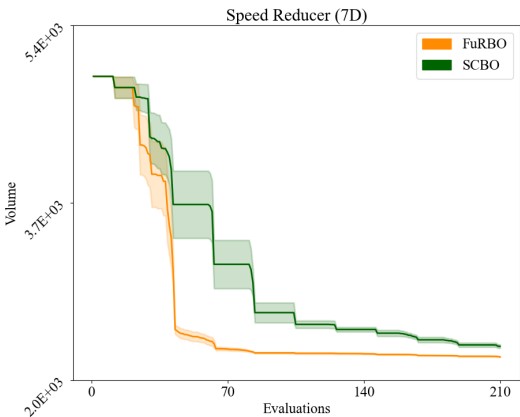

Figure 17: Convergence of FuRBO and SCBO on the speed reducer design problem with seven design parameters and eleven constraints. Results are averaged over 10 independent runs for each algorithm. The plot shows the mean objective value (volume), with shaded areas representing one standard error.

The objective function $f(x)$ is defined as a reward function, which combines the trajectory cost with penalties on initial and final positions and is given by:

$$f(x) = c(x) + \lambda \left( \|x_{0,1} - s\|_1 + \|x_{59,60} - g\|_1 \right) + b,$$

where $c(x)$ is a trajectory cost that penalizes any collision with an object along the trajectory by -20, $s$ and $g$ are the start and goal coordinates, $\lambda = -10$ is a weighting factor, and $b = 5$ is a bias term.

We redefine the hard constraint functions $c_i(x)$, each associated with a red obstacle $o_i$, as:

$$c_i(x) = \begin{cases} \displaystyle\sum_{\alpha \in \gamma(x) \cap o_i} d(\alpha, \text{center}(o_i)) & \text{if } \gamma(x) \cap o_i \neq \emptyset, \\ -\min_{\alpha \in \gamma(x)} d(\alpha, \text{center}(o_i)) & \text{otherwise,} \end{cases}$$

where $d(\cdot, \cdot)$ denotes the Euclidean distance.

This defines a maximization problem in which the rover is penalized for traversing yellow obstacles and for deviating from the start and goal positions. The 15 red obstacles are enforced as hard constraints: if the rover intersects a red square, the corresponding constraint returns the total distance from the center to the intruding trajectory points; otherwise, it returns the negative distance from the center to the closest trajectory point. The constraint $c_i(x) \leq 0$ ensures that a trajectory solution avoids the $i$-th red obstacle.

For performance evaluation, both SCBO and FuRBO are initialized with 100 samples, use a batch size of 100, and are run for a total budget of 2000 evaluations. As shown in Figure 18b, SCBO's convergence stagnates at a lower reward value compared to FuRBO.

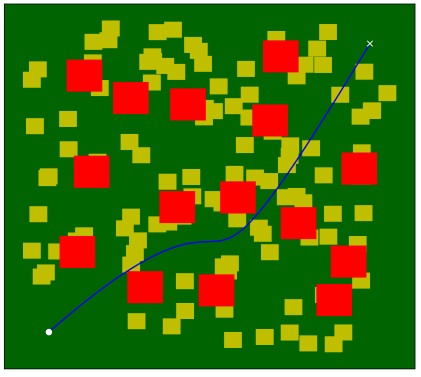

(a) Trajectory planning problem.

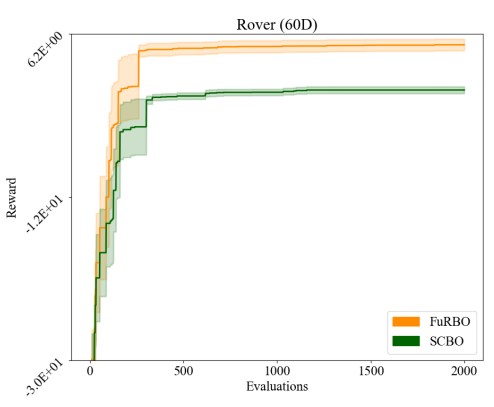

(b) Convergence.

Figure 18: Study on the trajectory-finding problem of a rover in dimension 60, under 15 constraints. (a) Visualization of the design domain problem and a trajectory solution. The yellow squares represent the obstacles the rover can overcome at a pre-fixed cost. The red squares represent the obstacles the rover must avoid (hard constraints). The blue line is a trajectory solution from the starting point (the white circle) to the goal point (the white cross). (b) Performance comparison between SCBO and FuRBO, averaged over 10 independent runs per algorithm.

