# OpenReview forum: "Feasibility-Driven Trust Region Bayesian Optimization"
_automl.cc/AutoML/2025/Methods_Track — AutoML 2025 Methods Track_

### Official Review · Reviewer_drYJ · 2025-04-29

**Comments To Authors:**

This paper proposes the FuRBO algorithm for high-dimensional Bayesian optimization (BO) with constraints. The proposed algorithm is an adaptation of the SCBO algorithm by Eriksson and Poloczek. Based on the argument that SCBO struggles to find initial feasible solutions, FuRBO uses a different technique to define the trust region (TR). In particular, the authors propose to sample points around the incumbent solution (if it is feasible, otherwise the point of lowest maximum normalized constraint violation) and rank those points according to a ranking metric. The smallest hyperrectangle including those best-performing points according to that metric defines the TR.

Strengths:
- The paper is well-written, and the problem is clearly motivated.
- The algorithm is empirically evaluated on various benchmark functions, showing that it is as good as SCBO or better on many benchmarks.
- The authors provide ablation studies for many important parameters, such as $P$, the number of initial DoE points, and the batch size

Weaknesses:
- The authors say that their algorithm is better than SCBO at finding feasible solutions if those are difficult to find, but do not give an explanation or intuition of why that is.
- Why is the batch size $q$ chosen to be three. There are ablations for the batch size, but I wonder why you did not use $q=1$ in the main text.
- Some design decisions are not well-motivated. For example, on p.5, line 192, why use $\max$ and not, for example, the mean?
- The overall contribution is relatively small and the method iterates over existing methods.

Minor comments:
- Page 2, lines 57-61 are hard to follow and unclear. It sounds like the authors are somehow sampling multiple _distributions_ while, in fact, they're sampling from distributions.
- Typo: 'isocontor', line 55, p.2
- Fig 1: the font size is too small
- page 5: the $\lceil P\cdot N\rceil$ does not work since $0\leq P \leq 100$
- Algorithm 1: It is not clearly stated why some lines are yellow
- The choice of the failure and success tolerances follows TuRBO. In some places, TuRBO could be cited more explicitly.


My main problem with this paper is that a clear motivation for why the improved TR definition works better if feasible solutions are hard to find is missing.

**Review Confidence:**

4

**Review Rating:**

6

---

### Official Review · Reviewer_iVsa · 2025-04-30

**Comments To Authors:**

Summary Of Contributions:
The authors present a Feasibility-Driven Trust Region Bayesian algorithm that iteratively defines a trust region from which the next candidate is selected, using information from both the objective and constraint surrogate models. The trust region shifts and resizes significantly between iterations, enabling the focus on its search and consistently accelerate the discovery of feasible and good-quality solutions. We empirically demonstrate the performance on the full BBOB-constrained COCO benchmark suite, comparing the state-of-the-art across varying levels of constraint severity and problem dimensionalities.

Potential Impact On The Field:
The proposed method would have potential good impacts on the field as the work promises to reach feasible zones faster than compared algorithms. Potentially increase the search performance. It can be used in AutoML literature, black-box optimization related application fields such as protein design, engineering, etc.

Technical Quality, Correctness, and Clarity:
The technical quality of the paper is good and looks structured with correct notations. Also, the proposed algorithm and its performance are explained clearly.

Suggestions:
I would recommend some improvements as follows in case of publication:
1. Related work structure should be improved and up-to-date publications would enhance the quality of the chapter and the publication.
2. Problem Formulation chapter looks like a definition more than a problem formulation. I would recommend that a proper problem be formalised with definitions of constraint region, trust region, feasibility check and what is the research question is being looked for.
3. Sometimes acronyms are between the brackets and sometimes the opposite. It cuts the consistency, therefore, I would recommend using one of them. Also, the authors should check the repetitive acronym usage, for instance, FurBO is defined more than once.

Ethics And Accessibility:
There is no ethical considerations and the results seems reproducible.

Overall Review:
Accept

**Review Confidence:**

4

**Review Rating:**

8

---

### Official Review · Reviewer_NWUF · 2025-04-30

**Comments To Authors:**

## Summary:
This paper introduces a new algorithm for constrained BO, FuRBO, that enhances existing trust region approaches by making it "feasibility-driven".
It builds directly upon SCBO, which centers its search region based on the best evaluated point.
FuRBO constructs its trust region by sampling and ranking so-called "inspector points" using both objective and constraint surrogate models loosely related to ideas from sampling based evolutionary strategies.
The trust region is then dynamically adjusted in shape and position to improve exploration of narrow, hard-to-identify feasible regions.
The method is evaluated on the BBOB constrained benchmark suite and compared against SCBO and other baselines.
Results show that FuRBO’s performs well, especially in high-dimensional and heavily constrained benchmark scenarios.

## Strengths:
Problem statement and motivation are clear and the related work and SCBO are explained well, and how and why existing trust region approaches in constrained BO might not scale to higher dimensions and many constraints.

Clarity is high, and the paper is written well.

The usage of the so-called inspectors to adapt the trust region as well as adaption of the trust region and the usage of a ranking mechanism intuitively makes sense.

In the benchmark experiments, FuRBO performs well, especially in high-dimensional problems and many constraints and benchmark design and analysis appears to be solid (although no real-world problems are benchmarked but solely the BBOB constrained benchmark suite).

The paper includes ablations studies regarding FuRBO's hyperparameter inspector percentage and with respect to the size of the initial design and batch size in the context of batch BO.

## Weaknesses:

FuRBO introduces additional computational overhead due to the inspectors (which - depending on the cost of the black box evaluations - can be relevant or might be irrelevant).
Still, runtime analyses in the appendix show that this might not be too bad for FuRBO.

Not all hyperparameters of FuRBO are exhaustively ablated (number of inspectors, top-percentage, initial variance, success/failure thresholds) and it is not fully clear how to best set them.

Novelty per se is slightly limited, as FuRBO is directly inspired by SCBO and solely differs in the trust region mechanism (but this should not be considered a strong weakness).

As with previous trust region approaches, the trust region is still restricted to be an axis-aligned hyperrectangle.

Experiments are restricted to synthetic BBOB constrained problems only.

## Ethical and Reproducibility Concerns:
There are no ethical issues.

Code is provided and well organized, with clear instructions how to reproduce results and analyze results and create figures.

## Rating:

The paper presents an interesting and empirically solid performing contribution to constrained BO.
Moderate impact and relevance to the AutoML community is given.
Overall, I therefore vote for acceptance.

**Review Confidence:**

3

**Review Rating:**

7

---

### Official Review · Reviewer_FtPZ · 2025-05-01

**Comments To Authors:**

This work introduces FuRBO (Feasibility-Driven Trust Region Bayesian Optimization), a novel algorithm for high-dimensional constrained black-box optimization problems. FuRBO uses adaptive trust regions to efficiently identify feasible, high-quality solutions. Empirical results on the BBOB-constrained COCO benchmark suite show that FuRBO outperforms state-of-the-art methods across a range of constraint severity and dimensions (2–40).

FuRBO extends SCBO (Scalable Constrained Bayesian Optimization) but differs in its trust region construction. Rather than applying Thompson sampling directly within a hypercube, FuRBO first generates an *inspector population* and ranks them using trained surrogate models for both the objective and constraints. It then selects the top-ranking inspectors to form a hyperrectangle, from which the batch of candidate solutions is selected using Thompson sampling. This design allows FuRBO to more heavily factor in constraint feasibility, which is particularly valuable in high-dimensional constrained settings.

I appreciate the novelty of the proposed algorithm but have several questions and suggestions:

- **Line 199**: It is unclear what “based on the evaluated models” means. Are inspector candidates ranked based on the posterior mean of the models? If so, I am wondering how applying Thompson sampling at this step would influence performance.
- **Line 189**: For infeasible points, the ranking is based on the maximum normalized constraint violation. I wonder whether using *total constraint violation* (as in SCBO) would be more effective in encouraging constraint minimization?
- It is interesting that **cEI** performs well in the cigar problem and early stages of other problems but tends to fall into local optima. I wonder if **cKG** could address this issue. A benchmark comparison with cKG would be useful, especially since it is cited in the literature review.
- For clarity, I suggest **swapping Figures 2 and 4**, as Figure 4 conveys the same concept more clearly.
- It is somewhat surprising that **cEI's** CPU time is comparable to SCBO and FuRBO, both of which use Thompson Sampling. Some explanation of the AF maximization setup would help clarify this.

Overall, the manuscript is well written and makes a good contribution to high-dimensional constrained Bayesian optimization. I recommend acceptance, but encourage the authors to carefully consider the suggestions above in revision.

**Review Confidence:**

4

**Review Rating:**

8

---

### Meta-Review · Area_Chair_81fc · 2025-05-01

**Recommendation:** Accept
**Confidence:** 4

**Metareview:**

The paper proposes FuRBO, a Bayesian optimization method for high dimensional black box functions with black box constraints. The algorithm refines an earlier approach, SCBO, and achieves strong empirical results.

I anticipate that the method will be useful in practice and spark further research in this important domain. The work would benefit from elaborating the intuition why FuRBO works better in highly constrained settings and from more ablation studies. I recommend that the authors address this feedback in their revision.

The reviews are generally favorable and emphasize the expected usefulness of FurBO for the community. Thus, I recommend accepting the submission to AutoML.